# Astrocytes regulate brain extracellular pH via a neuronal activity-dependent bicarbonate shuttle

Shefeeq M. Theparambil[1,4], Patrick S. Hosford[1,4], Iván Ruminot[2], Olga Kopach [3], James R. Reynolds [3], Pamela Y. Sandoval [2], Dmitri A. Rusakov [3], L. Felipe Barros [2] & Alexander V. Gourine [1✉]

Brain cells continuously produce and release protons into the extracellular space, with the rate of acid production corresponding to the levels of neuronal activity and metabolism. Efficient buffering and removal of excess $H^+$ is essential for brain function, not least because all the electrogenic and biochemical machinery of synaptic transmission is highly sensitive to changes in pH. Here, we describe an astroglial mechanism that contributes to the protection of the brain milieu from acidification. In vivo and in vitro experiments conducted in rodent models show that at least one third of all astrocytes release bicarbonate to buffer extracellular $H^+$ loads associated with increases in neuronal activity. The underlying signalling mechanism involves activity-dependent release of ATP triggering bicarbonate secretion by astrocytes via activation of metabotropic $P2Y_1$ receptors, recruitment of phospholipase C, release of $Ca^{2+}$ from the internal stores, and facilitated outward $HCO_3^-$ transport by the electrogenic sodium bicarbonate cotransporter 1, NBCe1. These results show that astrocytes maintain local brain extracellular pH homeostasis via a neuronal activity-dependent release of bicarbonate. The data provide evidence of another important metabolic housekeeping function of these glial cells.

[1] Centre for Cardiovascular and Metabolic Neuroscience, Neuroscience, Physiology and Pharmacology, University College London, London, UK. [2] Centro de Estudios Científicos (CECs), Valdivia, Chile. [3] Institute of Neurology, University College London, London, UK. [4] These authors contributed equally: Shefeeq M. Theparambil, Patrick S. Hosford. ✉email: a.gourine@ucl.ac.uk

Maintaining pH homeostasis is fundamentally important for uninterrupted activity of individual neurons and effective communication within neuronal circuits, which makes the brain information processing possible. Neurons constantly produce and release into the extracellular space significant amounts of acid equivalents, with the rate of acid production corresponding to the levels of neuronal activity and energy usage. Any failure to effectively counteract extracellular acid loads would compromise the function of neuronal circuits, simply because all the electrogenic and biochemical machinery of synaptic transmission is highly sensitive to pH[1-7].

Buffering by $CO_2/HCO_3^-$ is one of the most important mechanisms of tissue pH control. In this system, $CO_2$ and $H_2O$ are in a dynamic equilibrium with $H^+$ and $HCO_3^-$. This equilibrium is rapidly attained by the activity of enzymes from the carbonic anhydrase family[8-10]. Bicarbonate buffering and carbonic anhydrase activity protect brain tissue from acidification by converting $H^+$ and $HCO_3^-$ to $H_2O$ and $CO_2$. $CO_2$ is then removed by cerebral circulation and transported to the lungs to be exhaled. However, little is known about the mechanisms that maintain local $CO_2/HCO_3^-$ buffer strength in the extracellular space of the brain. Indeed, the brain extracellular space occupies only ~20% of the tissue volume[11,12], and the efficacy of the $CO_2/HCO_3^-$ buffering system would rapidly decline if $HCO_3^-$ is depleted in conditions of significant extracellular acid loads, such as during periods of increased neuronal activity. Therefore, maintaining stability of brain tissue pH necessitates an effective mechanism capable of supplying $HCO_3^-$ to the extracellular space in a responsive mode, i.e., in a neuronal activity-dependent manner.

Amongst different cellular players within the synaptic neuropil, astrocytes appear to be well suited to provide active control of local brain extracellular pH ($pH_e$) microenvironment. A single astrocyte occupies a large volume of brain parenchyma with an extensive arborisation covering thousands of individual synapses[13]. This anatomical arrangement enables astrocytes to monitor local brain activity by sensing neuronal signalling molecules (such as glutamate and ATP) that escape from the synaptic cleft[14-17]. Astrocytes characteristically express a high level of electrogenic sodium-bicarbonate cotransporter 1 (NBCe1, *Slc4a4*)[18,19] which is a high affinity carrier primarily responsible for transporting $HCO_3^-$ across the astroglial membrane[20,21]. NBCe1 activity is modulated by intracellular signalling mechanisms involving $Ca^{2+}$, cyclic adenosine monophosphate/protein kinase A, and phospholipase C (PLC)[22].

We hypothesised that, in astrocytes, the recruitment of one (or several) of these intracellular mechanisms in response to neuronal signalling molecules can stimulate outward activity of NBCe1 and thus supply $HCO_3^-$ to the extracellular space 'on demand', and in accord with the level of local neuronal activity. To test this hypothesis, we examined the effects of purinoceptor activation on $HCO_3^-$ transport and intracellular pH ($pH_i$) regulation in astrocytes, investigated the cellular mechanisms of $HCO_3^-$ release, and determined the effect of NBCe1 deletion in astrocytes on brain pH regulation. The results obtained in this study suggest that bicarbonate transport in astrocytes of the forebrain is controlled by purinergic signalling. ATP and downstream purines facilitate $HCO_3^-$ release by astrocytes via PLC/$Ca^{2+}$-mediated activation of NBCe1. This astroglial mechanism appears to play an important role in the control of local brain extracellular pH.

## Results

### Regulation of brain extracellular pH. 
In the absence of carbonic anhydrase catalytic activity, the rate of $H^+$ removal could be too slow to effectively counteract extracellular acid loads associated with brain activity. Indeed, inhibition of carbonic anhydrase with acetazolamide leads to a reversible extracellular acidification (by $-0.07 \pm 0.02$ pH units) of brain tissue (somatosensory cortex), as is evident from a significant decrease in the pH-sensitive electrochemical current recorded in the vicinity of the acetazolamide microinjection site (Fig. 1a). Profound and sustained extracellular acidification was also recorded in the brain after systemic administration of acetazolamide (Supplementary Fig. S1a). This result indicates that brain cells continuously produce and extrude $H^+$ into the extracellular space. It also implies that $H^+$ buffering by $HCO_3^-$ and carbonic anhydrase activity that facilitates the reversible conversion of $H^+$ and $HCO_3^-$ to $H_2O$ and $CO_2$ are essential for the maintenance of constant brain extracellular pH.

In the mouse somatosensory cortex, pH measurements with carbon fibre microelectrodes (CFM), coupled with fast scan cyclic voltammetry, showed that during periods of increased neuronal activity the extracellular pH remains unchanged or shifts in the alkaline direction during or immediately after the stimulation (regardless of the duration of the somatosensory stimulation) (Fig. 1b; Supplementary Fig. S1b). Effective buffering of the extracellular $H^+$ loads would require a supply of extra $HCO_3^-$, since extracellular $HCO_3^-$ pools are readily depletable while $H^+$ generation scales with the neuronal activity and energy use[23,24]. Using two-photon excitation imaging of cortical astrocytes loaded with a pH-sensitive dye BCECF (Fig. 1c), we next studied astroglial $[H^+]_i$ responses to the increases in local neuronal activity (Fig. 1d, e). On average, the population of cortical astrocytes responded to the activation of somatosensory pathways with intracellular acidification ($p < 0.001$). Strong intracellular acidification (change in BCECF fluorescence $\geq$ 2 standard deviations from the mean) was recorded in 27% of cortical astrocytes ($n = 32/117$ cells; Fig. 1d, e) and alkalinisation in 4% of astrocytes ($n = 5/117$ cells; Fig. 1d). The specificity of the recorded $pH_i$ responses was confirmed by giving the animals 10% $CO_2$ to breathe. As expected, $CO_2$ inhalation triggered intracellular acidification in the majority (63%, $n = 60/95$ cells) of the recorded cortical astrocytes (Supplementary Fig. S1c, d).

In acute hippocampal slices, electrical stimulation of Schaffer collateral fibres triggered intracellular acidification in 53% of recorded CA1 astrocytes ($n = 53/100$ cells; Fig. 2a–d) and alkalinisation in 30% of astrocytes ($n = 30/100$ cells; Fig. 2c,d). In baseline conditions, no significant fluctuations or drift of $pH_i$ values were detected over comparable recording periods (Fig. 2c). The magnitude of $pH_i$ changes recorded in CA1 astrocytes was dependent on the strength of the Schaffer collateral recruitment with significant $pH_i$ transients observed in response to a single pulse electrical stimulation (Fig. 2c, d). The proportion of astrocytes responding with changes in $pH_i$ to increases in the neuronal activity recorded in vitro was larger compared to that recorded in vivo, likely reflecting differences in tissue perfusion between the preparations (superfusion from the surfaces of the slice vs perfusion with the blood flow in vivo). Together, the data obtained in vivo and in vitro show that between 30–50% of all astrocytes respond to the increases in local neuronal activity with intracellular acidification. As extracellular pH is well maintained during periods of increased neuronal activity these data suggest that a significant proportion of astrocytes release $HCO_3^-$ to counteract the activity-associated extracellular acid loads (Fig. 1f).

**ATP modulates bicarbonate transport in astrocytes.** Emerging evidence suggests that communication between neurons and astrocytes is mediated primarily by purinergic signalling. ATP is released together with glutamate at central synapses[25-27], and astrocytes are well equipped with purinoceptors to sense synaptic activity by responding to changes in extracellular ATP[28].

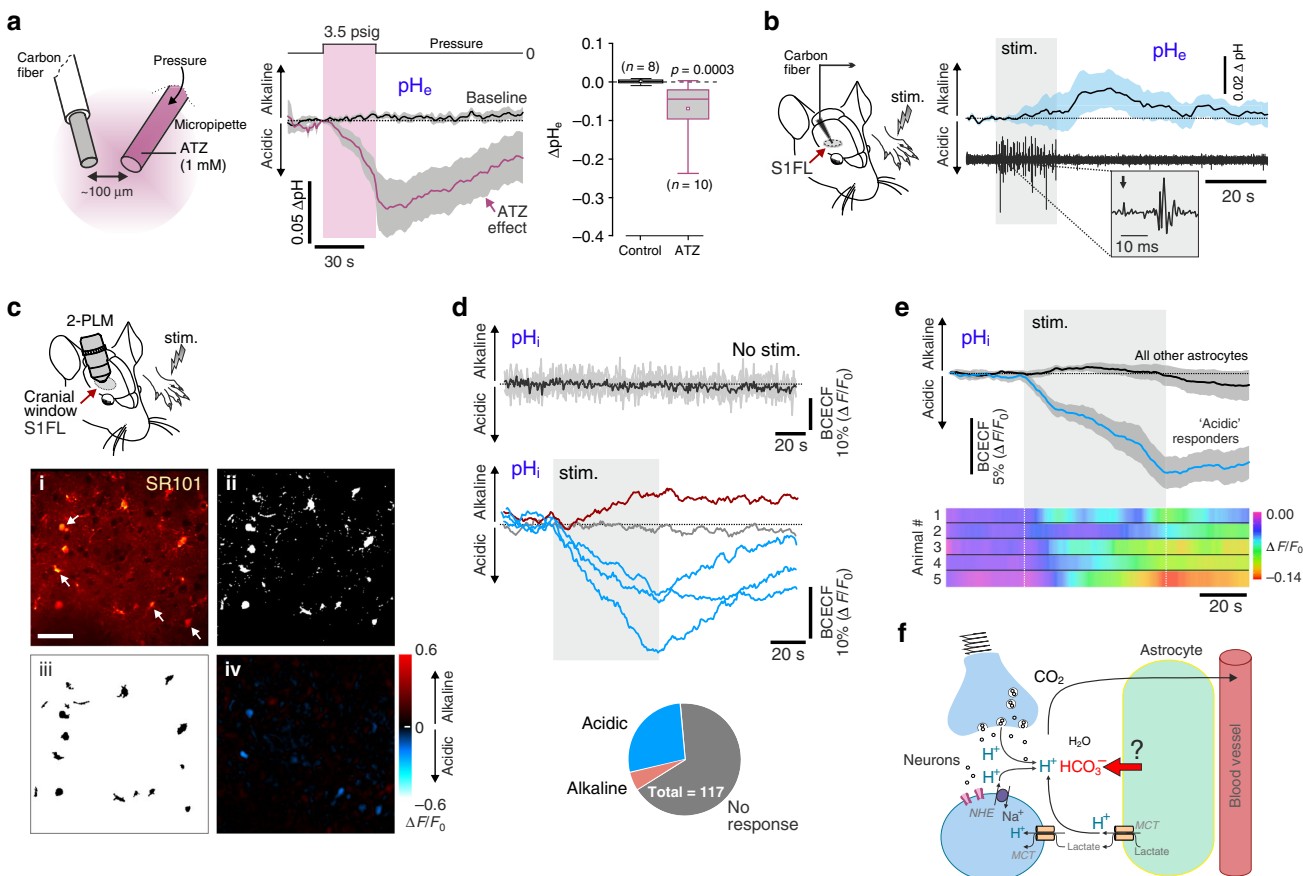

**Fig. 1 Regulation of brain extracellular pH. a** The role of the carbonic anhydrase: the effect of carbonic anhydrase inhibitor acetazolamide (ATZ) on extracellular pH ($pH_e$) in the somatosensory cortex of anaesthetised rats. Microinjection of ATZ (1 mM, placed ~100 μm from the tip of the recording electrode) leads to a reversible decrease in brain $pH_e$ as evident from a reduction in pH-sensitive electrochemical current recoded by fast cyclic voltammetry. Traces illustrate averaged (means ± SEM) changes in pH-sensitive current. Box-and-whisker plot illustrates peak changes in $pH_e$ in response to ATZ: the central dot indicates the mean, the central line indicates the median, the box limits indicate the upper and lower quartiles, and the whiskers show the minimum–maximum range of the data. *p* value, Mann–Whitney *U* test. **b** No extracellular acidification in response to increased neuronal activity and post-stimulus alkalinisation: Time course of $pH_e$ changes in the right forelimb region of the somatosensory cortex (S1FL) induced by activation of somatosensory pathways (electrical stimulation of the contralateral paw; 3 Hz, 1.5 mA, 20 s) in anaesthetised mice (*n* = 13). Changes in $pH_e$ and neuronal responses were recorded simultaneously using carbon fibre microelectrodes. Traces illustrate averaged (means ± SEM) changes in pH-sensitive current and representative recordings of extracellular potential in response to somatosensory stimulation. **c** In vivo imaging of intracellular pH ($pH_i$) responses in astrocytes of the S1FL cortex using two-photon laser scanning microscopy (2-PLM) in mice. (i) Cortical astrocytes loaded with a pH sensitive dye BCECF and identified by sulphorhodamine (SR101) labelling (arrows). Scale bar = 50 μm. (ii) SR101 staining was used to binarise the image, whereby (iii) astrocyte cell bodies were segmented as contiguous SR101-positive regions. All identified astrocytes were included in the analysis. (iv) A representative example of peak BCECF fluorescence changes recorded in S1FL cortical astrocytes in response to the electrical stimulation of the contralateral paw. **d** Representative examples of changes in BCECF fluorescence in five S1FL cortical astrocytes induced by activation of somatosensory pathways. Pie chart shows the proportion of astrocytes responding to increased neuronal activity with intracellular acidification and alkalinisation recorded in vivo. **e** Neuronal activity-induced intracellular acidification in astrocytes suggesting that astroglia is a source of bicarbonate: Time course of $pH_i$ changes recorded in S1FL cortical astrocytes in response to electrical stimulation of the contralateral paw. Traces illustrate averaged (means ± SEM) changes in BCECF fluorescence recorded in cortical astrocytes that showed peak change in $\Delta F/F_0 \geq 2$ SD of baseline fluorescence (responders; 27% of the whole population) and all other astrocytes. False colour plots illustrate averaged changes in BCECF fluorescence in all SR101-labelled cells in each individual animal. **f** Schematic drawing of the neurovascular unit illustrating the sources of extracellular $H^+$. Acid loads associated with neuronal activity are hypothesised to be buffered by $HCO_3^-$ derived from astrocytes. NHE, sodium hydrogen exchanger. MCT, monocarboxylate transporter. Source data are provided as a Source Data file.

Extracellular ATP concentration increases during neuronal activation[29], and mediates the neurovascular coupling response at the capillary level[29–31]. Therefore, we next tested the hypothesis that $HCO_3^-$ transport in astrocytes is modulated by purinergic signalling.

In a bicarbonate buffered solution (26 mM $[HCO_3^-]_e$ saturated with 5% $CO_2$), cultured cortical astrocytes responded to ATP with strong intracellular acidification (increase in $[H^+]_i$ by 18 ± 1 nM and 66 ± 4 nM in response to 200 μM and 1 mM ATP, respectively; *p* < 0.001) (Fig. 3a), which followed elevations in

$[Ca^{2+}]_i$ (see below and Supplementary Fig. S2a). ADP (200 μM) had the same effect (increase in $[H^+]_i$ by 33 ± 2 nM, *p* < 0.001; Fig. 3c; Supplementary Fig. S2b). Neither ATP nor ADP had an effect on intracellular pH in conditions when $CO_2/HCO_3^-$-buffer was replaced with bicarbonate-free solution (HEPES buffer saturated with $O_2$) (Fig. 3b, d), suggesting that the ATP/ADP-induced intracellular acidification in astrocytes develops as a result of facilitated outward $HCO_3^-$ transport.

The ATP/ADP-induced decreases in intracellular pH in astrocytes were markedly reduced in the presence of pharmacological

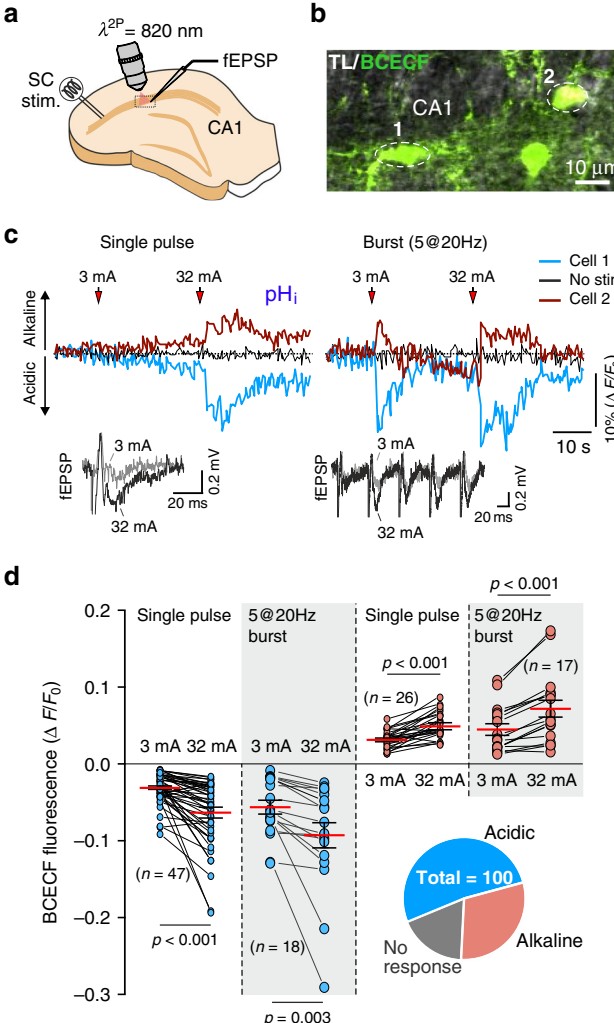

**Fig. 2 Neuronal activity-dependent intracellular pH responses in hippocampal astrocytes in vitro. a** Schematic drawing of the experimental design. Electrical stimulation of Schaffer collateral (SC) fibres was applied during two-photon imaging of $pH_i$ in hippocampal astrocytes and recordings of field excitatory postsynaptic potentials (fEPSP). **b** Representative image illustrating astrocyte cell bodies and process loaded with BCECF and surrounding CA1 pyramidal cells (not stained). **c** Representative examples of changes in BCECF fluorescence in two CA1 astrocytes (shown in **b**) induced by electrical stimulation of Schaffer collateral fibres. The stimulations were applied in two modes: a single pulse mode with stimuli of 3 or 32 mA in amplitude or burst stimulation with a train of 5 pulses of 3 or 32 mA in amplitude, each applied at 20 Hz. Insets illustrate representative examples of the recorded fEPSPs. **d** Summary data illustrating peak changes in BCECF fluorescence induced in CA1 astrocytes by Schaffer collateral fibre stimulation of increasing strength. Pie chart shows the proportion of astrocytes responding to increases in neuronal activity with intracellular acidification and alkalinisation recorded in these experiments. Data are presented as individual values and means ± SEM. Numbers in parentheses indicate the numbers of individual astrocytes recorded in 5–9 acute hippocampal slice preparations obtained from at least five animals. *p* values, two-sided paired *t*-test. Source data are provided as a Source Data file.

agents that inhibit $Na^+/HCO_3^-$ cotransporter (NBC) activity (S0859, 50 and 100 μM; 4,4′-Diisothiocyano-2,2′-stilbenedisulfonic acid, DIDS, 200 μM) (Fig. 3e, f; Supplementary Fig. S2c, d), and abolished in conditions of genetic NBCe1 deletion (Fig. 3g, h). Membrane depolarisation by raising the extracellular $[K^+]$ (from 3

to 7 mM and then to 20 mM) reduced the magnitude of ATP-induced intracellular acidification in astrocytes (Supplementary Fig. 3a–d). This result is consistent with the properties of NBCe1 and shows that outwardly directed transport of bicarbonate by NBCe1 is thermodynamically less favourable at the membrane potentials less negative than the NBCe1 equilibrium potential (calculated at −71 mV). Collectively, these data suggest that purinergic signalling modulates NBCe1 activity in astrocytes, with ATP (and downstream purines) acting to facilitate the outward transport of $HCO_3^-$.

This conclusion was also supported by measurements of intracellular $[Na^+]$. NBCe1 cotransports $HCO_3^-$ and $Na^+$. Therefore, in response to ATP, the direction of intracellular $[Na^+]$ change is determined by the NBCe1-mediated outward $Na^+$ transport, by $Na^+$ entry via enhanced sodium–calcium exchange (secondary to $Ca^{2+}$ responses), and by activation of ionotropic P2X receptors (all occurring in parallel with the background activity of $Na^+/K^+$ ATPase)[32]. We found that the balance of these ATP-induced actions results in a net increase in $[Na^+]_i$ (Supplementary Fig. S2e). Bicarbonate-free conditions enhanced the amplitude of ATP-induced $[Na^+]_i$ responses (Supplementary Fig. S2e, f). Because it is predominantly the $Na^+/HCO_3^-$ cotransport that is affected in the absence of $HCO_3^-$, this result is consistent with the facilitation of outward NBCe1 activity in response to ATP.

**Expression of NBCe1 in the cerebral cortex**. The single-cell RNAseq data from the mouse cerebral cortex (Fig. 4a) demonstrated high expression of gene encoding NBCe1 (*Slc4a4*) in discrete populations (or clusters) of cells (Fig. 4b). The analysis of cell identity marker genes showed that these clusters express characteristic astrocyte-specific genes (Fig. 4c). *Slc1a3* (glutamate aspartate transporter 1) and *Gja1* (connexin 43) are strongly and uniformly expressed across all cell clusters that display high expression of *Slc4a4* and are negative for the expression of genes associated with neurons (*Syt1*), pericytes (*Kcnj8*), smooth-muscle cells (*Acta2*), and microglial cells (*Tmem119*) (Fig. 4c). We, therefore, refer to these cells as astrocyte marker-positive cells. The analysis of RNAseq data from these cells taken in isolation (6,760 cells) revealed 11 sub-clusters of astrocyte marker-positive cells with distinct RNA expression profiles (Fig. 4d). *Slc4a4* and the gene encoding carbonic anhydrase 2 (*Car2*) showed significant differential expression within these cell clusters (Fig. 4e). A large cluster comprised of >1000 cells (Cluster 2) displayed high expression of both *Slc4a4* and *Car2* genes (average logFC of 0.52; adj. $p = 1.4 \times 10^{-61}$, and 0.26; adj. $p = 1.2 \times 10^{-20}$, respectively). The proportion of cells within the clusters that were *Slc4a4*-positive and/or *Car2*-positive, and the relative average expression of these genes (Fig. 4e), showed the heterogeneity of *Slc4a4* and *Car2* expression in astrocyte marker-positive cells. Clusters 1 and 2 contained the largest proportion of cells expressing high levels of both *Slc4a4* and *Car2*, accounting for more than 30% of all astrocyte marker-positive cells (Fig. 4f). These data indicate that in the cerebral cortex NBCe1 is expressed predominantly in astrocytes, and that in a significant proportion of these cells high NBCe1 expression is complemented by high expression of carbonic anhydrase 2.

**NBC mediates activity-dependent $pH_i$ responses in astrocytes**. In the experiments in acute hippocampal slices we next tested the hypothesis that NBC is responsible for the neuronal activity-dependent $pH_i$ changes in astrocytes. Pharmacological blockade of NBC activity with S0859 (50 μM) or DIDS (300 μM) markedly reduced or abolished (depending on the strength of the stimulus) $pH_i$ responses in CA1 astrocytes triggered by the electrical

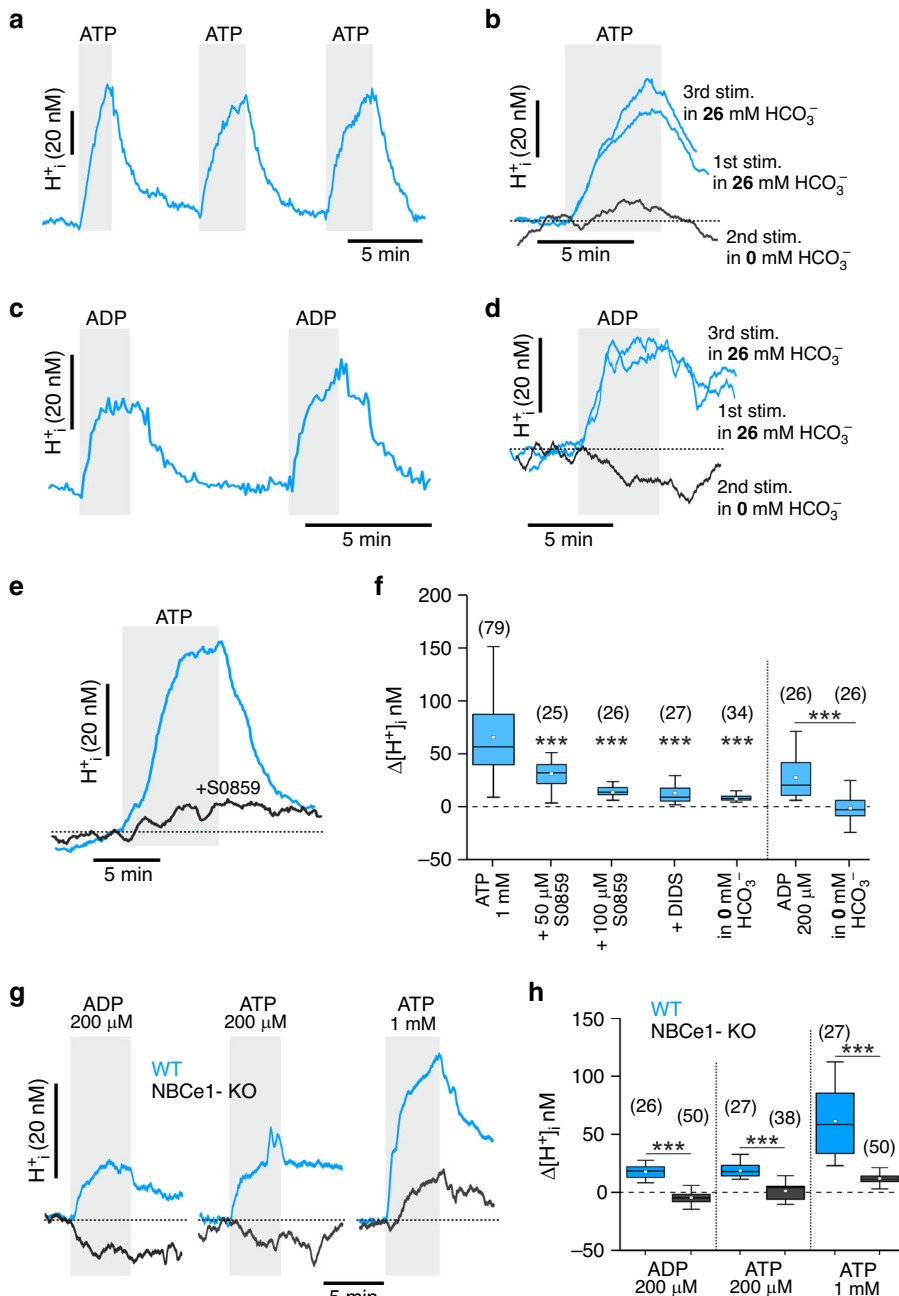

**Fig. 3 Bicarbonate transport in astrocytes is modulated by purinergic signalling. a** Representative example of intracellular acidification induced by ATP in cortical astrocytes in culture. The trace depicts averaged $[H^+]_i$ responses of 17 astrocytes to repeated application of ATP (1 mM) in culture. **b** ATP-induced intracellular acidification in astrocytes is reversibly blocked by removal of extracellular $HCO_3^-$. Averaged traces of ATP-induced $[H^+]_i$ responses in 9 astrocytes in the presence and absence of $HCO_3^-$ in the media are shown. **c** Representative example of intracellular acidification induced by ADP in cortical astrocytes in culture. The trace depicts averaged $[H^+]_i$ responses of 9 astrocytes to repeated application of ADP (200 μM) in culture. **d** ADP-induced intracellular acidification in astrocytes is reversibly blocked by removal of extracellular $HCO_3^-$. Averaged traces of ADP-induced $[H^+]_i$ responses in 14 astrocytes in the presence and absence of $HCO_3^-$ in the media are shown. **e** ATP-induced intracellular acidification in astrocytes is blocked by sodium bicarbonate co-transporter (NBC) inhibitor S0859 (100 μM). Averaged traces of ATP-induced $[H^+]_i$ responses in 12 astrocytes are shown. **f** Summary data illustrating the effects of pharmacological inhibition of NBC (S0859, DIDS) or removal of extracellular $HCO_3^-$ on peak increases in $[H^+]_i$ induced by ATP or ADP in astrocytes. **g** Representative examples of averaged $[H^+]_i$ responses to ADP or ATP in 12 cultured astrocytes of wild-type (WT) and 12 astrocytes of electrogenic sodium-bicarbonate cotransporter 1 (NBCe1) knockout (NBCe1-KO) mice. **h** Summary data illustrating peak changes in $[H^+]_i$ induced by ATP and ADP in astrocytes of WT and NBCe1-KO mice. In the box-and-whisker plots the central dot indicates the mean, the central line indicates the median, the box limits indicate the upper and lower quartiles and the whiskers extend to 1.5 IQR from the quartiles. Numbers in parentheses indicate the numbers of individual astrocytes recorded in 3–5 separate cultures prepared from the same number of animals. *** denotes difference from the response recorded in control conditions (**f**) or in WT astrocytes (**h**) with $p < 0.001$ (one-way ANOVA with Tukey's post-hoc test). Source data are provided as a Source Data file.

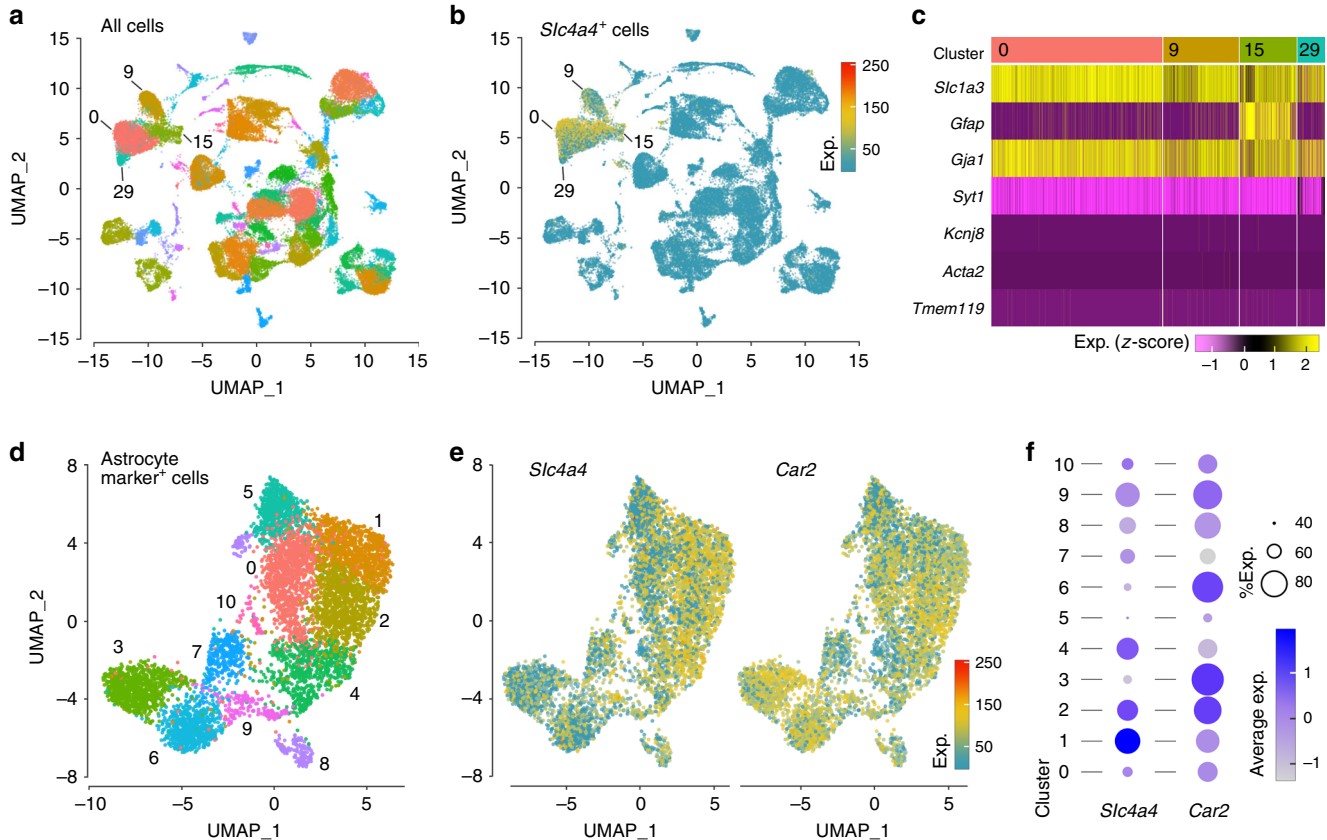

**Fig. 4 NBCe1 gene *Slc4a4* is expressed predominantly in astrocytes. a** Visualisation of cerebral cortex single-cell mRNA sequencing data from the Mouse Brain Atlas database[54] using Uniform Manifold Approximation and Projection (UMAP). Each colour represents a 'cluster' of cells that share similarities in RNA expression profile. Expression data from a total of 49,703 cells were analysed yielding 63 distinct clusters. **b** Distribution of *Slc4a4* expression across all the identified clusters of cortical cells. Cells within the four labelled clusters display the highest differential expression of *Slc4a4*, illustrated as reads per million of RNA molecules detected. **c** *Slc4a4*-positive cell clusters show high level of expression of astrocyte-specific marker genes. Heatmap showing scaled (z-score) cell marker gene expression in selected cell clusters: 0, 9, 15, and 29. Genes indicated are astrocyte-associated: *Slc1a3*, glutamate aspartate transporter 1 (GLAST-1); *GFAP*, glial fibrillary acidic protein, *Gja1*, connexin 43. Also analysed are neuron-associated gene: *Syt1*, synaptotagmin-1; pericyte-associated gene: *Kcnj8*, Kir6.1; smooth-muscle cell-associated gene: *Acta2*, α-smooth-muscle actin; and microglia-associated gene: *Tmem119*, transmembrane protein 119. Magenta colour indicates low gene expression, yellow colour indicates high gene expression. **d** Visualisation and re-clustering of cells identified as astrocytes using UMAP at high resolution. Expression data from a total of 6760 cells showing high expression of astrocyte-specific gene markers were analysed and yielded 11 distinct astroglial sub-clusters based on similarities in the RNA expression profile. **e** Expression profile of *Slc4a4* and carbonic anhydrase 2 (*Car2*) in cortical cell clusters identified as astrocytes. Expression levels are illustrated as reads per million of RNA molecules detected. **f** Relative expression of *Slc4a4* and *Car2* genes within the astroglial sub-clusters. Percentage of cells within each cluster that are positive for *Slc4a4* and *Car2* is represented by the size of the circle and the degree of the relative expression is represented by the colour gradient. Circles representing clusters with a high proportion of cells that strongly express the gene of interest appear larger and darker blue.

stimulation of Schaffer collateral fibres (Fig. 5). Activity-dependent pH$_i$ changes in individual CA1 astrocytes responding with intracellular acidification (Fig. 5a–c) and intracellular alkalinisation (Fig. 5d, e) were reduced by the NBC blockade. These data are consistent with the evidence that NBCe1 in astrocytes can transport bicarbonate in both directions[19,33]. These data also argue against a significant role of other potential cytosolic acid loading mechanisms, such as Cl$^-$/HCO$_3^-$ exchange and proton uptake via glutamate transporters in the mechanisms underlying the neuronal activity-dependent intracellular acidification in astrocytes (conclusion also supported by the results of our earlier reports[21,33]).

Patch-clamp recordings in CA1 astrocytes (nine cells recorded in acute slice preparations from four animals, resting membrane potential below −75 mV) showed that pharmacological NBC inhibition leads to a reversible hyperpolarization. S0859 (50 μM, 10 min application) reduced the median membrane potential of CA1 astrocytes from −80.7 to −85.0 mV ($p = 0.004$; paired-sample Wilcoxon test; Supplementary Fig. 3e,f). These data

provide evidence that under resting conditions, NBCe1 in astrocytes is operating in the outward mode.

**Signalling mechanisms of NBCe1 activity modulation by ATP.** We next investigated the signalling mechanisms responsible for the activation of the NBCe1-mediated outward transport of HCO$_3^-$ in response to ATP. ATP-induced intracellular acidification and [Ca$^{2+}$]$_i$ responses were markedly reduced in the presence of the broad spectrum P2 receptor antagonist PPADS (100 μM) (Fig. 6a, b). However, neither TNP-ATP (10 μM), which preferentially inhibits ionotropic P2X receptors, nor the P2X$_7$ antagonist O-ATP (100 μM) had an effect on ATP-induced pH and Ca$^{2+}$ responses in astrocytes (Supplementary Fig. S4a–d).

Extracellular ATP is rapidly broken down by ectonucleotidase activity to ADP that is predominantly active at metabotropic P2Y receptors. That the effects of ATP on intracellular pH and [Ca$^{2+}$]$_i$ in astrocytes are mimicked by ADP (Fig. 3) suggests that metabotropic P2Y receptors mediate these effects. Indeed, the

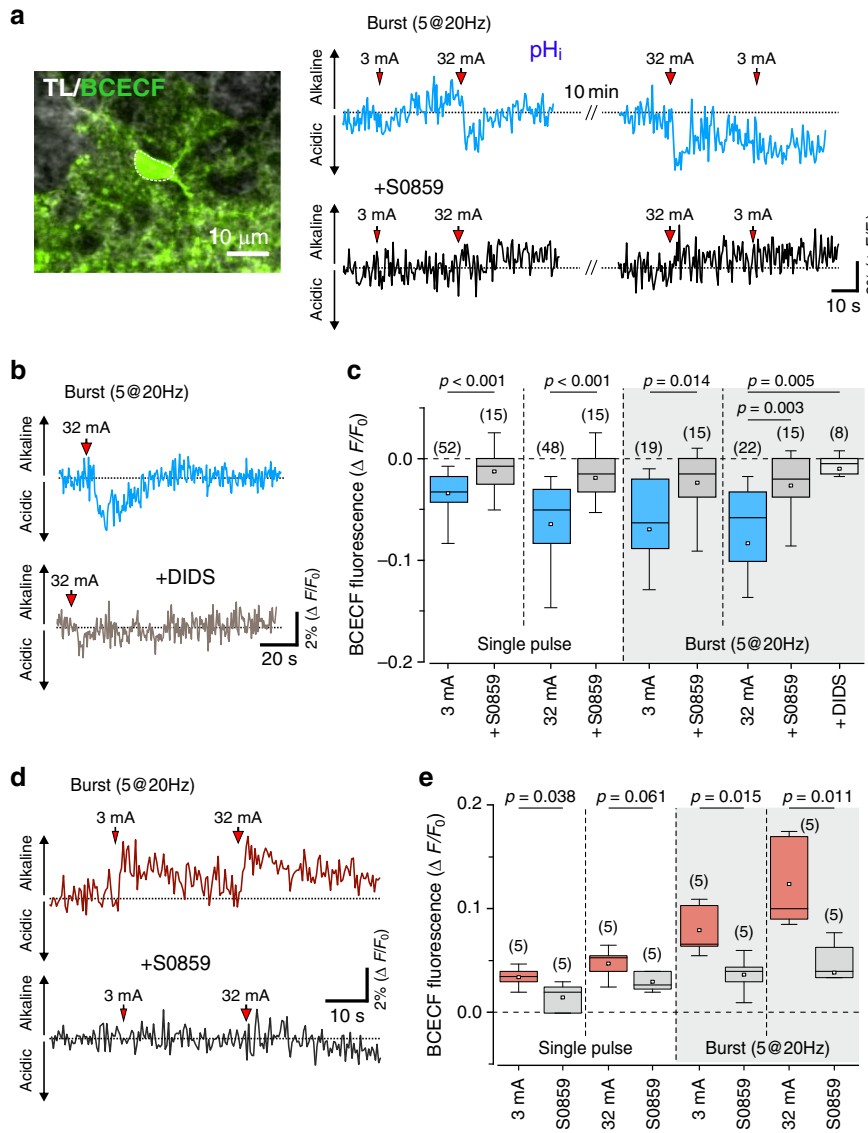

**Fig. 5 Neuronal activity-dependent intracellular pH responses in astrocytes are mediated by NBC. a** Representative example of changes in BCECF fluorescence indicative of intracellular acidification in a single CA1 astrocyte induced by electrical stimulation of Schaffer collateral fibres in the absence and presence of the NBC blocker S0859 (50 µM). **b** Representative example of changes in BCECF fluorescence in CA1 astrocyte induced by Schaffer collateral stimulation in the absence and presence of DIDS (300 µM). **c** Summary data illustrating the effects of pharmacological inhibition of NBC (S0859, DIDS) on peak decreases in BCECF fluorescence induced in CA1 astrocytes by the electrical stimulation of Schaffer collateral fibres. **d** Representative example of changes in BCECF fluorescence indicative of intracellular alkalinisation in a CA1 astrocyte induced by Schaffer collateral stimulation in the absence and presence of S0859. **e** Summary data illustrating the effect of S0859 on peak increases in BCECF fluorescence induced in CA1 astrocytes by the electrical stimulation of Schaffer collateral fibres. In the box-and-whisker plots the central dot indicates the mean, the central line indicates the median, the box limits indicate the upper and lower quartiles and the whiskers extend to 1.5 IQR from the quartiles. Numbers in parentheses indicate the numbers of individual astrocytes recorded in 5–9 acute hippocampal slice preparations obtained from at least four animals. *p* values, one-way ANOVA. Source data are provided as a Source Data file.

blockade of P2Y$_1$ receptors with MRS2179 (20 µM) or MRS2500 (2 µM) reduced or abolished the ATP or ADP-induced intracellular acidification and [Ca$^{2+}$]$_i$ responses (Fig. 6c, g, h; Supplementary Fig. S5c). The specific P2Y$_1$ receptor agonist 2-Mes-ADP (20 µM) induced intracellular acidification and Ca$^{2+}$ responses in astrocytes (Fig. 6d, g, h; Supplementary Fig. S5b). These responses were similar to the responses induced by ATP/ADP and were abolished by the P2Y$_1$ receptor blocker MRS2500 (Fig. 6d, g, h).

An increase in [Na$^+$]$_i$ following activation of a sodium–calcium exchanger is one of the potential mechanisms that can facilitate the outward NBCe1 activity[34]. However, the blockade of

sodium–calcium exchange with SN-6 (20 µM) had no effect on ATP-induced intracellular pH and [Ca$^{2+}$] responses in astrocytes (Supplementary Fig. S4e,f). There is also evidence that PLC-mediated Ca$^{2+}$ recruitment from the intracellular stores can modulate NBCe1 activity[22]. Activation of P2Y$_1$ receptors in astrocytes recruits PLC activity[28,35]. ATP/ADP effects on intracellular pH and [Ca$^{2+}$] in astrocytes were found to be abolished when PLC activity was blocked with edelfosine (10 µM) or U73122 (10 µM) or when intracellular Ca$^{2+}$ stores were depleted by thapsigargin (1 µM) (Fig. 6e–h, Supplementary Fig. 5a, d). These data suggest that astrocytes release bicarbonate to the extracellular space in response to ATP/ADP acting at

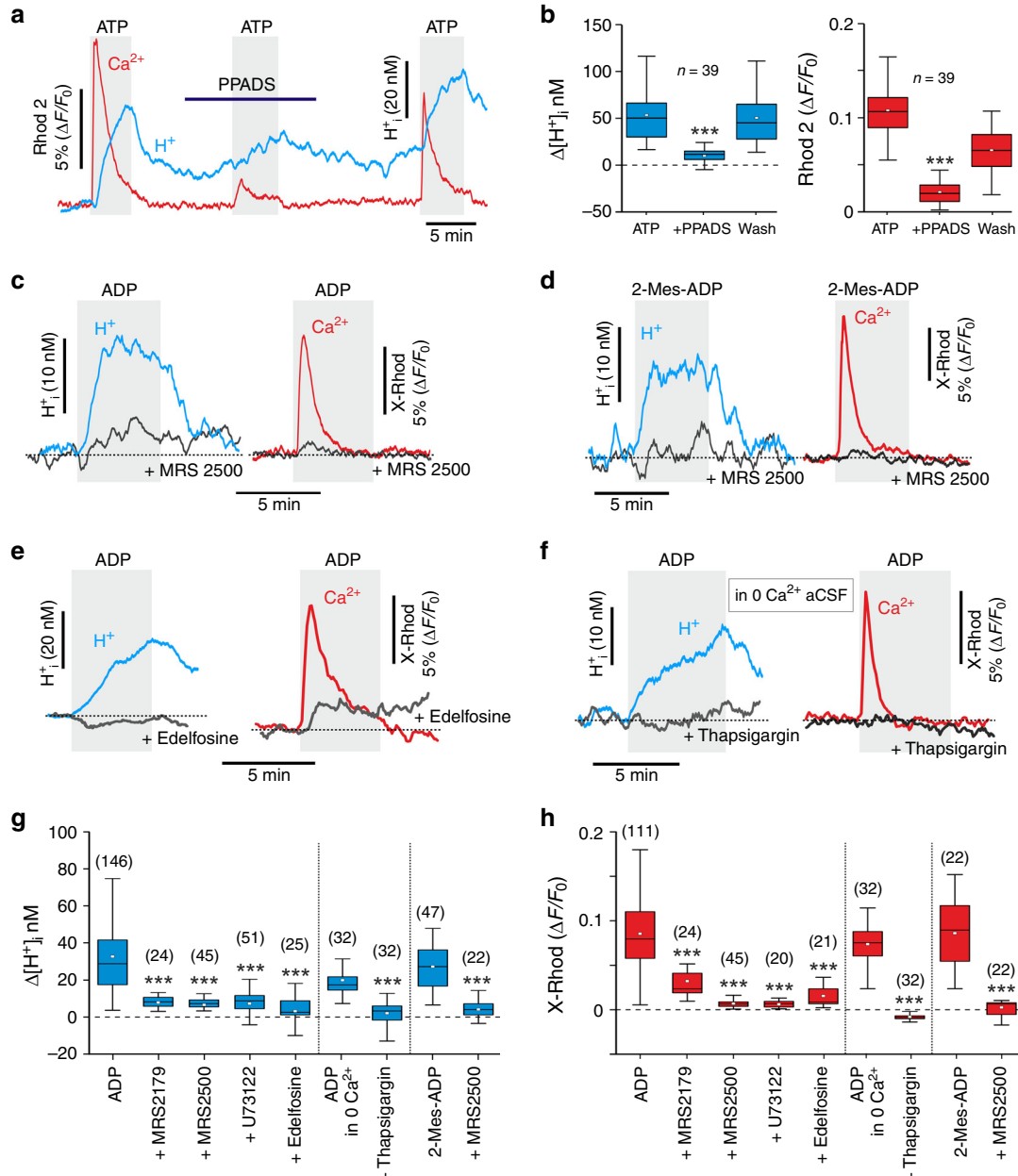

**Fig. 6 Signalling mechanisms underlying purinergic modulation of bicarbonate transport in astrocytes. a** Representative recording illustrating the effect of P2 receptor antagonist PPADS (100 μM) on $[H^+]_i$ and $[Ca^{2+}]_i$ responses induced by ATP (1 mM) in astrocytes. Averaged traces of $[H^+]_i$ and $[Ca^{2+}]_i$ changes in ten astrocytes are shown. **b** Summary data illustrating the effects of PPADS on peak increases in $[H^+]_i$ and $[Ca^{2+}]_i$ induced by ATP in astrocytes. **c** Intracellular acidification and $Ca^{2+}$ responses in astrocytes induced by ADP (200 μM) are blocked by $P2Y_1$ receptor antagonist MRS2500 (2 μM). Averaged traces of ADP-induced $[H^+]_i$ and $[Ca^{2+}]_i$ responses in ten astrocytes are shown. **d** The effects of ATP/ADP on $[H^+]_i$ and $[Ca^{2+}]_i$ in astrocytes are mimicked by $P2Y_1$ receptor agonist 2-Mes-ADP (20 μM). Averaged traces of 2-Mes-ADP-induced $[H^+]_i$ and $[Ca^{2+}]_i$ responses in the absence and presence of MRS2500 in 13 astrocytes are shown; **e** Intracellular acidification and $[Ca^{2+}]_i$ responses in astrocytes induced by ADP are blocked by phospholipase C inhibitor edelfosine (10 μM). Averaged traces of ADP-induced $[H^+]_i$ and $[Ca^{2+}]_i$ responses in 11 astrocytes are shown. **f** Intracellular acidification and $[Ca^{2+}]_i$ responses in astrocytes induced by ADP are blocked after depletion of intracellular $Ca^{2+}$ stores with thapsigargin (1 μM) in the absence of extracellular $Ca^{2+}$. Averaged traces of ADP-induced $[H^+]_i$ and $[Ca^{2+}]_i$ responses in 16 astrocytes are shown. **g, h** Summary data illustrating the effects of $P2Y_1$ receptor blockade (MRS2179, 20 μM; MRS2500, 2 μM), phospholipase C inhibition (U73122, 10 μM; edelfosine, 10 μM), removal of extracellular $Ca^{2+}$ (0 $Ca^{2+}$ conditions) or removal of extracellular $Ca^{2+}$ combined with depletion of intracellular $Ca^{2+}$ stores with thapsigargin (1 μM) on peak increases in $[H^+]_i$ and $[Ca^{2+}]_i$ induced by ADP (200 μM) in astrocytes. Peak $[H^+]_i$ and $[Ca^{2+}]_i$ responses induced in astrocytes by 2-Mes-ADP (20 μM) in the absence and presence of MRS2500 are also shown. In the box-and-whisker plots, the central dot indicates the mean, the central line indicates the median, the box limits indicate the upper and lower quartiles and the whiskers extend to 1.5 IQR from the quartiles. Numbers in parentheses indicate the numbers of individual astrocytes recorded in 3–5 separate cultures prepared from the same number of animals. *** denotes difference from the response recorded in control conditions with $p < 0.001$ (one-way ANOVA with Tukey's post-hoc test). Source data are provided as a Source Data file.

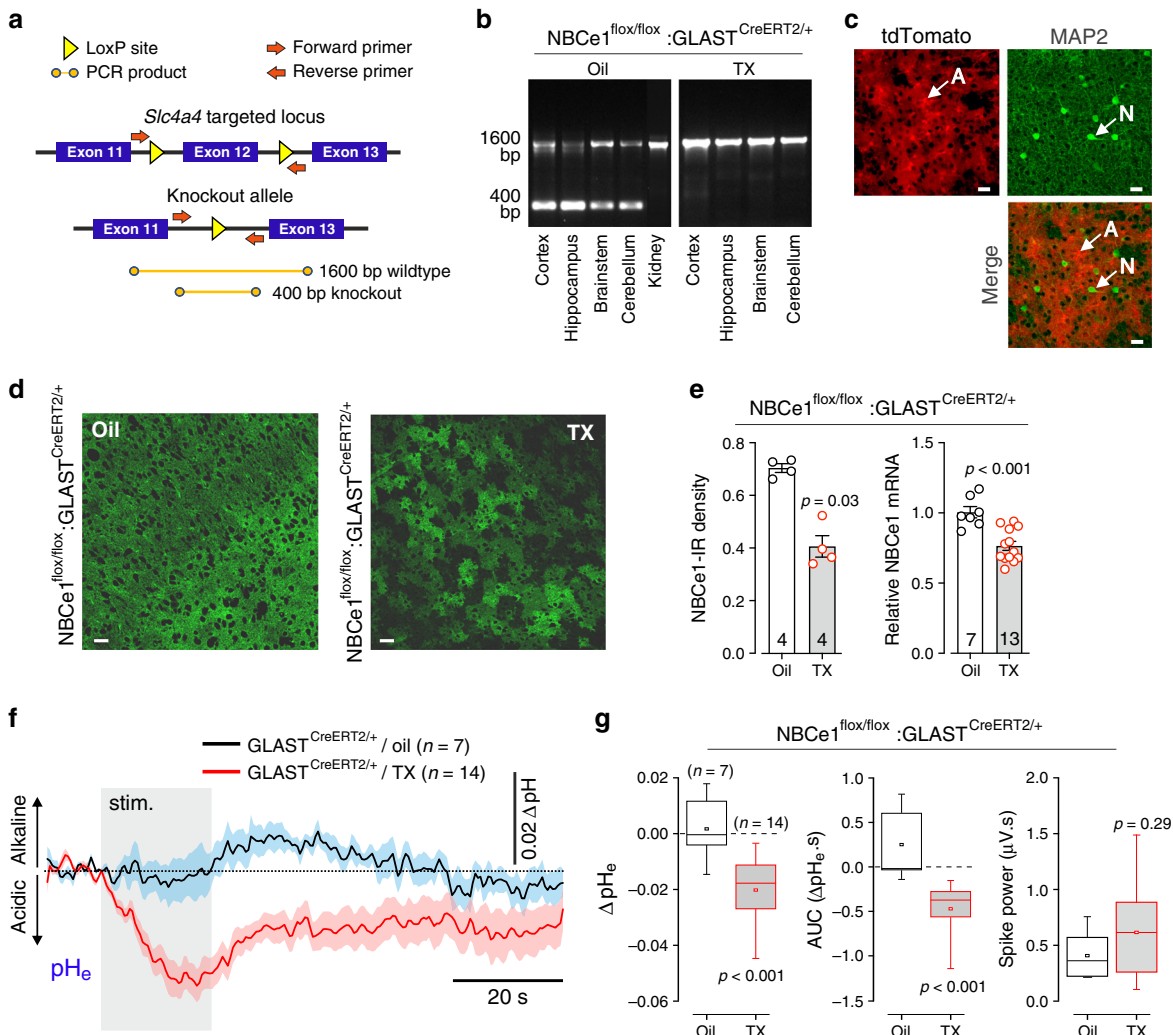

**Fig. 7 Astroglial NBCe1 is essential for the regulation of brain extracellular pH. a** Genotyping strategy used to determine the efficacy of recombination and deletion of exon 12 of *Slc4a4* locus. The expected wild-type and gene knockout products are 1600 bp and 400 bp, respectively. **b** PCR-products showing the recombination status of *Slc4a4* locus in different brain regions of vehicle (oil) and tamoxifen (TX)-treated NBCe1$^{flox/flox}$/GLAST$^{CreERT2/+}$ mice. Wild-type fragment is present across all the samples while the knockout band is present only in the brain tissue samples confirming specific induction of recombination in astrocytes. **c** Mosaic expression of the reporter gene (tdTomato) in protoplasmic cortical astrocytes (A). Neurons (N) were identified by MAP2 immunostaining. Scale bars = 20 μm. **d** Knockdown of NBCe1 expression in cortical astrocytes. Confocal images illustrate immunohistochemical detection of NBCe1 in the cortex of NBCe1$^{flox/flox}$:GLAST$^{CreERT2/+}$ mice treated with the vehicle or tamoxifen. Tamoxifen treatment of NBCe1$^{flox/flox}$:GLAST$^{CreERT2/+}$ mice resulted in a mosaic pattern of NBCe1 expression. Scale bars = 20 μm. **e** Quantification of tamoxifen effect on cortical NBCe1 expression (NBCe1 immunostaining density and relative NBCe1 mRNA levels) in NBCe1$^{flox/flox}$:GLAST$^{CreERT2/+}$ mice. NBCe1-IR, NBCe1 immunoreactivity. Data are presented as individual values and means ± SEM. **f** Time course of extracellular pH changes in the right S1FL cortical region induced by activation of somatosensory pathways (electrical stimulation of the contralateral paw; 3 Hz, 1.5 mA, 20 s) in NBCe1$^{flox/flox}$:GLAST$^{CreERT2/+}$ mice treated with the vehicle (oil) or tamoxifen. Traces illustrate averaged (means ± SEM) changes in pH-sensitive current. In conditions of NBCe1 knockdown in astrocytes, somatosensory stimulation resulted in a significant extracellular acidification in the S1FL cortex. **g** Summary data illustrating peak and integral (area under the curve, AUC) changes in pH$_e$ and neuronal activity (expressed as spike power) evoked in the S1FL region by electrical forepaw stimulation in NBCe1$^{flox/flox}$:GLAST$^{CreERT2/+}$ mice treated with the vehicle or tamoxifen. In the box-and-whisker plots the central dot indicates the mean, the central line indicates the median, the box limits indicate the upper and lower quartiles and the whiskers show the minimum-maximum range of the data. *p* values, Mann–Whitney *U* test. Source data are provided as a Source Data file.

metabotropic P2Y$_1$ receptors, through the downstream activation of PLC, the recruitment of Ca$^{2+}$ from the intracellular stores, and the facilitation of HCO$_3^-$ outward transport by NBCe1.

**Astroglial NBCe1 is essential for the control of brain pHe**. To understand the adaptive significance of the astroglial mechanism of NBCe1-mediated HCO$_3^-$ release found in this study, we next used the NBCe1 conditional knockdown strategy to reduce the expression of this transporter specifically in astrocytes (Fig. 7a–e).

Tamoxifen treatment of NBCe1$^{flox/flox}$:GLAST$^{CreERT2/+}$ mice reduced the NBCe1 transcript level by ~25% and the density of NBCe1 immunostaining by ~40% (Fig. 7e), resulting in a mosaic pattern of NBCe1 expression in the cortex (Fig. 7d). The degree of NBCe1 knockdown with this approach was smaller than expected. Yet, even in conditions of partial NBCe1 knockdown in astrocytes, extracellular pH was not maintained, and significant extracellular acidification developed in the somatosensory cortex in response to electrical stimulation of the forepaw (Fig. 7f–h). These data suggest that astrocytes actively release HCO$_3^-$ via

NBCe1 to counteract extracellular acid loads associated with increases in local neuronal activity.

## Discussion

The results obtained in this study show that ATP and ADP (the first product of extracellular ATP breakdown) trigger bicarbonate secretion by astrocytes through the activation of metabotropic P2Y$_1$ receptors, recruitment of PLC, release of Ca$^{2+}$ from the internal stores, and facilitated outward HCO$_3^-$ transport by NBCe1. Because enhanced neuronal/synaptic activity is commonly associated with increases in the local concentration of extracellular ATP[27,29,30] and with [Ca$^{2+}$]$_i$ elevations in neighbouring astrocytes[36–38], this mechanism is proposed to be responsible for the neuronal activity-dependent supply of bicarbonate to the extracellular space. This process is essential for the maintenance of extracellular pH buffer strength and, therefore, the brain pH homeostasis.

The brain milieu could be prone to significant fluctuations in pH due to the activities of several cellular and membrane mechanisms that involve the generation of acid/base equivalents and their movements across the cellular membranes. These mechanisms include metabolic production of CO$_2$/H$^+$, mono-carboxylate transporter-mediated co-transport of H$^+$ and lactate by neurons and glial cells, exocytosis of highly acidic neurotransmitter vesicles, the activity of plasma membrane Ca$^{2+}$ ATPase, bicarbonate flux via anion channels, transporters and GABA$_A$ receptors, and operation of H$^+$ extruders, such as Na$^+$/H$^+$ exchanger and V-type ATPase[8,23,39] (Fig. 1f). The balance between these processes leads to extracellular acid loads, since protons and CO$_2$ generated by active neurons and other brain cells must be removed if brain function is to be maintained. Yet, the brain extracellular pH remains remarkably stable, not only withstanding variable levels in neuronal activity and metabolism, but also major (physiological or pathological) perturbations of systemic acid/base balance[40].

The profile of activity-induced changes in extracellular pH we recorded in the mouse somatosensory cortex is broadly consistent with the pH$_e$ responses to physiological stimuli recorded in vivo and reported in earlier publications, in response to increased neuronal activity extracellular pH becomes more alkaline or remains unchanged[41–43]. Differences in experimental conditions, cerebral perfusion, species-specific organisation of afferent pathways, and other factors may contribute to the differences in the pH$_e$ responses to neuronal activity encountered in the literature. Importantly, these differences have no material impact on the conclusions drawn from the in vivo recordings reported in this study, where we observed a clear difference between genotypes under identical experimental conditions. Strong extracellular acidification associated with increased neuronal activity in conditions of NBCe1 knockdown in astrocytes points to the critical importance of this transporter for the regulation of brain extracellular pH.

We propose that astroglial supply of bicarbonate via NBCe1 maintains the effectiveness of the extracellular CO$_2$/HCO$_3^-$ buffering system at variable levels of local neuronal activity. High expression of NBCe1 in astrocytes is complemented by equally high expression of intracellular carbonic anhydrase (type 2)[44], confirmed by the results of the present analysis (Fig. 4). This allows astrocytes to effectively 'scavenge' some of locally generated CO$_2$ in order to maintain the intracellular HCO$_3^-$ stores and supply HCO$_3^-$ to the extracellular space when required. Indeed, intracellular HCO$_3^-$ generation during hypercapnic challenges appears to be essential for NBCe1-mediated HCO$_3^-$ release in cortical astrocytes and in *Xenopus* oocytes co-expressing NBCe1 and carbonic anhydrase 2[33]. The present results suggest that

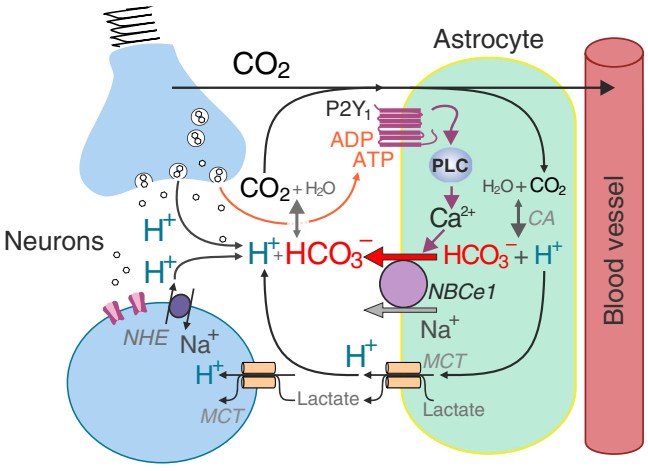

**Fig. 8 Brain CO$_2$/H$^+$ cycle.** Schematic drawing of the neurovascular unit illustrating the proposed cellular mechanisms contributing to the maintenance of brain extracellular pH homeostasis. We hypothesise that extracellular acid loads associated with enhanced neuronal activity are effectively buffered by HCO$_3^-$ released by astrocytes via the mechanism described in this study. CA, carbonic anhydrase, PLC, phospholipase C.

astrocytes in the forebrain are equipped with a mechanism that effectively counteracts extracellular acidification, via the NBCe1-mediated supply of bicarbonate 'on demand' and in accord with the prevailing levels of local neuronal activity and energy usage (Fig. 8).

The data obtained in this study also point to the existence of two distinct populations of astrocytes responding to local neuronal activity with the opposite changes in intracellular pH. A larger population constituting of up to 50% of all astrocytes respond with intracellular acidification, consistent with the activity-dependent outward transport of HCO$_3^-$ that prevents extracellular pH falls during periods of increased neuronal activity. A smaller population (~4% of cells recorded in vivo; 30% of cells recorded in vitro) respond with intracellular alkalinisation. Intracellular alkalinisation has been previously attributed to K$^+$-dependent HCO$_3^-$ uptake via the NBCe1[45–48], leading to glycolytic activation, reduction of oxygen consumption and fast lactate release by astrocytes[49–52]. Taken together, these results are consistent with the evidence suggesting that forebrain astrocytes are heterogeneous in terms of their resting membrane potential with a larger subpopulation having the membrane potential between −90 to −70 mV, and a 'depolarised' subpopulation with the membrane potential between −60 to −30 mV[53]. Considering that ion stoichiometry of astroglial NBCe1 is 1Na$^+$:2HCO$_3^-$ and that the equilibrium potential of NBCe1 is between −68 to −74 mV[21], this transporter would be expected to operate close to its reversal potential or in the outward mode in the majority of astrocytes. In a smaller subpopulation of astrocytes NBCe1 would operate in the inward mode.

In cortical astrocytes, a variety of extracellular acidic stimuli have been shown to trigger the rapid release of HCO$_3^-$ via NBCe1[33]. In this study, recruitment of polysynaptic somatosensory pathways in the in vivo experiments revealed the heterogeneous [H$^+$]$_i$ responses of astrocytes in the somatosensory cortex. We hypothesise that two populations of astrocytes fulfil distinct functional roles: a larger population that exports HCO$_3^-$and, by doing so, maintains extracellular pH homeostasis, and the second population in which inward NBCe1 activity facilitates glucose mobilisation and glycolysis, as described previously by one of our laboratories[49,50]. Analysis of single-cell RNAseq data of the mouse cerebral cortex from the Mouse Brain

Atlas database[54] identified 11 distinct clusters of cells based on similarities in the RNA expression profile, all expressing a high level of astrocyte-specific genes (Fig. 4). Although, NBCe1 expression is almost exclusively confined to astroglial population, it remains to be determined whether the cells that constitute different astrocyte sub-clusters based on their genetic makeup are also functionally distinct.

High levels of neuronal activity leading to significant elevations of extracellular $[K^+]$ may cause depolarisation of astrocytes and, therefore, would be expected to inhibit $HCO_3^-$ outward transport and favour NBCe1 operation in the inward mode. In pathological conditions like epilepsy, stroke, and spreading depression, when the brain extracellular $[K^+]$ markedly increases, the astroglial mechanism of extracellular pH control described in this study is likely to be disrupted. Indeed, early reports showed that in rats direct electrical stimulation of the brain tissue or evoked spreading depression trigger large depolarisations (up to +40 mV) and intracellular alkalinisation in cortical astrocytes leading to acidification of the extracellular space[46].

Importantly, under normal physiological conditions NBCe1-mediated transport is prone to modulation by key intracellular signalling pathways[22]. The activity of NBCe1 expressed in *Xenopus* oocytes has been shown to be facilitated in response to PIP2 degradation by PLC and intracellular $Ca^{2+}$ release[55]. There is also evidence that the increases in intracellular $Ca^{2+}$ and NBCe1 phosphorylation can potentially alter the ion stoichiometry of the transporter and, therefore, the direction of NBCe1-mediated bicarbonate transport[56,57]. The results of this study are consistent with the existing evidence that in the majority of astrocytes NBCe1 is operating in the outward mode and that the outward activity of the transporter is facilitated by P2Y$_1$ receptor activation via PLC recruitment and $Ca^{2+}$ release from the internal stores. We found no evidence that this signalling pathway changes NBCe1 stoichiometry or the direction of $HCO_3^-$ transport.

Interestingly, analogous mechanisms have been shown to operate in other physiological systems, although these may utilise other membrane transporters of bicarbonate. For example, in the gut epithelial cells, ATP acting at P2Y$_1$ receptors stimulates $HCO_3^-$ secretion via $Cl^-/HCO_3^-$ exchanger[58]. There is evidence for the existence of a local negative feedback mechanism that controls $HCO_3^-$ release by these cells with the key components including extracellular alkaline phosphatase, ATP, P2Y$_1$ receptors and membrane bicarbonate transport[58]. Akin to the mechanism described here, ATP stimulates $HCO_3^-$ secretion leading to alkalisation of the extracellular space. This optimises the activity of alkaline phosphatase and facilitates ATP degradation. The acidic environment inhibits alkaline phosphatase activity, prolongs ATP actions and promotes $HCO_3^-$ secretion[58]. We found no evidence of alkaline phosphatase involvement in the mechanism identified in this study (Supplementary Fig. 4g, h). Indeed, it seems advantageous for the extracellular concentration of ATP/ADP in the brain to be determined primarily by the level of local neuronal activity, and not by the tissue ectonucleotidase activity. There is also evidence that extracellular acidification by itself (independently of the synaptic activity) may facilitate the release of purines[59,60] and, therefore, potentiate bicarbonate release by the mechanism described here.

Because of the high pH sensitivity of neurotransmitter receptors, ion channels, and biochemical cascades involved in synaptic transmission, maintaining a stable pH environment is critically important for brain function. Respiratory chemoreceptors located in the brainstem restore brain pH in response to acute changes in the blood and brain $PCO_2$/pH by controlling the activity of the respiratory network leading to the adaptive changes in lung ventilation. There is evidence that brainstem astrocytes function as central respiratory chemoreceptors, sensitive to changes in $PCO_2$ and pH[61–65]. Interestingly, the mechanisms underlying the pH sensitivity of the brainstem astrocytes also involve NBCe1 and ATP-mediated signalling[61,63]. However, in contrast to cortical and hippocampal astrocytes, in the specialised chemosensory brainstem astrocytes NBCe1 operates in the inward mode[63]. In these astrocytes acidification leads to NBCe1-mediated increase in intracellular $[Na^+]$ which activates sodium/calcium exchanger to operate in a reverse mode, leading to increases in intracellular $[Ca^{2+}]$ and $Ca^{2+}$-dependent release of ATP[63,66]. Since brainstem astrocytes are adjacent to, and intermingled with, the neuronal respiratory control networks, they are in a position to directly modulate breathing activity (via the release of ATP[61]) and, therefore, maintain systemic (arterial) pH homeostasis. The data obtained in this study demonstrate the existence of another key astroglial mechanism, essential for the maintenance of local brain pH in face of variable extracellular acid loads that depend on neuronal activity.

Clinical studies have identified homozygous mutations of *Slc4a4* gene (leading to variable degree loss of NBCe1 function) causing permanent renal tubular acidosis, glaucoma and hemiplegic migraine[67,68]. Heterozygous carriers of *Slc4a4* mutation appear to be largely normal, but may display some of these pathological features[67]. Mouse models demonstrated that NBCe1 function is critically important for homeostasis as global *Slc4a4* knockout animals (with complete NBCe1 loss-of-function) do not survive beyond the third week of life[69]. Lifespan in these animals is limited by severe metabolic acidosis[69] where breathing deficits may contribute to this harmful phenotype, as discussed previously[63]. In this study, we employed an inducible astrocyte-specific NBCe1 knockout strategy whereby the animals develop normally. Behavioural studies focusing on the assessment of basic parameters such as stress, motor skills, balance, learning and memory in conditions of astrocyte-specific NBCe1 deletion are currently in progress.

Disturbances of brain pH homeostasis have been implicated in the pathogenesis of several common neurological conditions, including epilepsy, ischaemia/stroke, as well as different psychiatric disorders such as schizophrenia, bipolar disorder and autism spectrum disorders[70]. A strong correlation between full scale human intelligent quotient (IQ) and brain pH was reported in adolescents (7–13 years old boys), with lower pH associated with lower IQ scores[71]. Astrocytes control the extracellular concentrations of key ions (potassium in particular), and metabolites (lactate, glutamate/glutamine), therefore, are critically important for the maintenance of the ionic and metabolic homeostasis of the brain milieu. This study describes another important metabolic housekeeping role of these ubiquitous glial cells. The data show that astrocytes help to maintain the stability of local brain extracellular pH via a neuronal activity-dependent shuttle of bicarbonate.

## Methods

**Ethical approval and animal husbandry**. All animal experimentations were performed in accordance with the European Commission Directive 2010/63/EU (European Convention for the Protection of Vertebrate Animals used for Experimental and Other Scientific Purposes) and the UK Home Office (Scientific Procedures) Act (1986) with project approval from the Institutional Animal Care and Use Committees of the University College London and Centro de Estudios Científicos. The animals were group-housed and maintained on a 12-h light cycle (lights on 07:00) and had ad libitum access to water and food. The rats were housed in a temperature-controlled room at 22 °C with 55 ± 10% relative humidity. The mice were housed at 24 °C ambient temperature with relative humidity kept at 60 ± 5%.

**Animal preparation for the in vivo studies**. Young adult C57Bl/6J mice of both sexes (3–4 months old) were anaesthetised with either ketamine/xylazine (100 mg kg$^{-1}$/10 mg kg$^{-1}$) or a mix of fentanyl, medetomidine and midazolam (0.03 mg kg$^{-1}$/ 0.3 mg kg$^{-1}$/ 3 mg kg$^{-1}$, respectively). Rats (Sprague-Dawley,

250–300 g) were anaesthetised with α-chloralose (100 mg kg$^{-1}$). Adequate anaesthesia was ensured by maintaining stable levels of the heart rate showing lack of a response to a paw pinch. If required, supplemental doses of the anaesthetics were administered. The body temperature was maintained at 37 ± 1 °C. The animal was mounted in a stereotaxic frame. For the recordings of extracellular pH using fast scan cyclic voltammetry or intracellular pH using 2P microscopy, the right forelimb region of the somatosensory cortex (S1FL) was exposed following craniotomy (~4 mm in diameter) and removal of dura mater. To activate somatosensory pathways, unilateral electrical stimulation of the forepaw (300 μs pulse width, 3 Hz, 1.5 mA) was applied using bipolar subcutaneous electrodes delivering fixed-current pulses from an isolated stimulator (Digitimer DS3) controlled by a 1401 interface (Cambridge Electronic Design).

**Fast scan cyclic voltammetry in vivo**. Parenchymal pH and evoked neuronal activity in the S1FL region of the cortex were recorded using fast cyclic voltammetry[31]. Reference[43] gives detailed description of the technique, principles of H$^+$ detection, calibration, specificity and interference with the detection of other analytes. CFM (diameter 7 μm) were advanced into the S1FL cortex until evoked extracellular potentials were detected in response to the electrical stimulation (1 Hz) of the contralateral forepaw. A series of voltage ramps (200 V s$^{-1}$) from 0 to −1 V were applied to the CFM at a frequency of 2 Hz. The resulting current was amplified, digitised and recorded for offline isolation of faradaic currents corresponding to [H$^+$] changes. The CFM recordings were continuously switched between current amplification and voltage amplification to allow near-simultaneous detection of the evoked potentials (voltage) and electrochemical changes (current), proportional to changes in extracellular [H$^+$]. Trains of electrical forepaw stimulation were applied three times per animal/experiential condition with intervals of at least 3 min between the stimulations. Neuronal responses were analysed by integration of the evoked volley of extracellular potentials with the baseline noise subtracted. At the end of the recordings, pH sensitivity of each CFM was determined by calibration, as described[43].

**Two-photon imaging of pH$_i$ in astrocytes in vivo**. During the animal preparation, the exposed area of the cortex was superfused with warm artificial cerebrospinal fluid (aCSF; containing in mM; 125 NaCl, 3 KCl, 26 NaHCO$_3$, 1.25 Na$_2$HPO$_4$, 18 Glucose, 2 CaCl$_2$, 2 MgSO$_4$; saturated with 95% O$_2$/5% CO$_2$, pH 7.4). Cortical astrocytes were labelled with sulforhodamine 101 (SR101) and loaded with a pH sensitive dye BCECF. The solution containing BCECF (Invitrogen; 1 mM) and SR101 (5 μM) in aCSF was delivered via glass micropipettes at three separate sites within the S1FL region. The exposed surface of the cortex was then protected with a glass coverslip secured to the skull using acrylic dental cement.

Astroglial pH$_i$ responses induced by somatosensory stimulation in the S1FP region of the cortex were visualised via the cranial window using an Olympus FV1000 microscope (Olympus). Excitation illumination was generated by an Insight X3 DUAL laser (Spectra Physics) with a pulse rate of 80 MHz. A 25× water immersion objective (XLPlan N, NA 1.05, Olympus) was used. Fluorophores were excited in two-photon XYZ-t mode at 770 nm. Images were acquired up to 100–150 μm deep from the cortical surface. Z-stacks (~15 μm) were acquired and maximum-projected to enable tracking of cells that moved in the z plane during the experimental period.

Images were motion-corrected in XY using a fast Fourier transform method[72]. Astrocytes were identified by SR101 labelling, and SR101 emission-based binarisation was used for image segmentation to isolate astrocyte-specific pH$_i$ responses. A response threshold for pH changes was set at a two sigma change at the peak of the BCECF fluorescence response associated with the stimulus, compared to BCECF fluorescence at the baseline. Stimulus-evoked changes in astrocyte pH$_i$ were compared to baseline recordings where no stimulus was applied. A systemic CO$_2$ challenge (10% inspired CO$_2$), known to induce strong acidosis in the neuropil, was applied via a ventilation mask and used as a positive control.

**Hippocampal slice preparation**. Young Sprague-Dawley rats (P21-P25) or C57Bl/6J mice (~1 month old) were terminally anaesthetized with isoflurane, the brains were isolated and coronal hippocampal slices (350 μm) were cut in an ice-cold slicing solution containing (in mM): 64 NaCl, 2.5 KCl, 1.25 NaH$_2$PO$_4$, 0.5 CaCl$_2$, 7 MgCl$_2$, 25 NaHCO$_3$, 10 glucose and 120 sucrose, saturated with 95% O$_2$/5% CO$_2$. Slices were then left to recover for 1 h in aCSF containing in mM; 126 NaCl, 3 KCl, 26 NaHCO$_3$, 1.25 Na$_2$HPO$_4$, 10 Glucose, 2 CaCl$_2$, 2 MgSO$_4$, saturated with 95% O$_2$/5% CO$_2$ (pH 7.4).

**Two-photon imaging of pH$_i$ in astrocytes in vitro**. Slices were incubated in aCSF containing BCECF (5 μM) for 45 min followed by a through washing out to allow de-esterification of the dye. Optical recordings of changes in pH$_i$ were performed in a flow-through imaging chamber mounted on a stage of an Olympus FV1000 system optically linked to a Ti:Sapphire MaiTai laser with $\lambda^{2P}$ = 820 nm (Spectra Physics). Recordings were performed at ~33–35 °C in aCSF saturated with 95% O$_2$/5% CO$_2$ (pH 7.4; 300–310 mOsmol). Astroglial loading with BCECF was confirmed by analysing the characteristic morphology of the BCECF-stained cells featuring astrocytes, with no fluorescence detected in principal neurons (CA1–CA3 pyramidal neurons) (Fig. 2b). For time-lapse recordings of pH$_i$

changes, images of BCECF-loaded astrocytes were collected using 256 × 256-pixel frames in the stream acquisition mode. Changes in pH$_i$ are expressed as changes in BCECF fluorescence at the maximum of the fluorescent signal over the baseline ($\Delta F/F_0$). Control optical recordings were performed in the same time frame without Schaffer fibre stimulation applied.

**Electrophysiology in vitro**. Recordings of field excitatory postsynaptic potentials (fEPSP) were performed using glass electrodes (1–1.5 MΩ) placed in the *stratum radiatum*. The signal was amplified (Multipatch 700B) and processed using pClamp 10.2 software (Molecular Devices). To stimulate Schaffer collateral fibres, a concentric bipolar stimulating electrode was placed in the stratum radiatum ~200 μm from the stratum pyramidale. The stimulations were applied in two modes: a single pulse mode with stimuli of 3 or 32 mA in amplitude (100 μs) or burst stimulation with a train of 5 pulses of 3 or 32 mA each applied at 20 Hz.

For the whole-cell recordings of the *stratum radiatum* astrocytes, acute hippocampal slices were placed in a recording chamber mounted on a stage of Olympus BX51WI microscope equipped with a LUMPlanFI/IR 40 × 0.8 objective coupled to an infra-red DIC imaging system. Recordings of resting membrane potential were performed in current-clamp mode using a Multipatch 700B amplifier controlled by the pClamp 10.2 software. The recording electrodes (4.0–5.5 MΩ) were filled with a solution containing (in mM) 130 Cs-methylsulfonate, 2 KCl, 0.5 EGTA, 10 HEPES, 4 Mg-ATP, 0.5 Na-GTP; 0.2 spermine tetrahydrochloride (pH 7.2, ~290 mOsmol). Astrocytes were identified by their morphological features and electrophysiological properties ($V_m$ < −75 mV and low membrane resistance).

**Primary astrocyte cultures**. Primary cultures of cortical astrocytes were prepared from the brains of rat and mouse (global NBCe1 knockout and wild-type littermates) pups (P2-3 of either sex) as described[20,63,66]. The animals were euthanized by isoflurane overdose, the brains were removed, and the cortical regions were separated by dissection. After isolation, the cells were plated on poly-D-lysine-coated coverslips and maintained at 37 °C in a humidified atmosphere of 5% CO$_2$ and 95% air for a minimum of 10 days before the experiments.

**Imaging of pH$_i$, [Ca$^{2+}$]$_i$ and [Na$^+$]$_i$ in cultured astrocytes**. Optical recordings of changes in [H$^+$]$_i$ and [Ca$^{2+}$]$_i$ in cultured astrocytes were performed using an inverted epifluorescence Olympus microscope, equipped with a cooled CCD camera (Clara, Andor, Oxford Instruments)[63,66]. For simultaneous recordings of changes in [H$^+$]$_i$ and [Ca$^{2+}$]$_i$, the cells were loaded with BCECF (Invitrogen; 2 μM for 10 min incubation[20]) and X-Rhod1 or Rhod2 (Invitrogen; 5 μM for 10 min incubation). After incubation with the dyes, cultures were washed three times prior to the experiment. Recordings were performed in a custom-made flow-through imaging chamber at ~32 °C in aCSF saturated with 95% O$_2$/5% CO$_2$ (pH 7.4). The rate of chamber perfusion with aCSF was 1 ml min$^{-1}$. Changes in [H$^+$]$_i$ and [Ca$^{2+}$]$_i$ were monitored in individual cells using excitation light provided by a xenon arc lamp with the beam passing sequentially through a multi dichroic beam splitter (Chroma Technology) at 575/10, 490/10 and 440/10 nm (Cairn Research). Fluorescence emission was recorded at 535, 590, and 600 nm for BCECF, Rhod2 and X-Rod1 indicators, respectively. Changes in BCECF fluorescence were converted to intracellular proton concentration ([H$^+$]$_i$ in nM) using the Nigericin /K$^+$ calibration method[20].

Intracellular [Na$^+$] changes in astrocytes were recorded using a sodium-sensitive fluorescent dye Asante-sodium green (ANG-2-AM, Abcam; 10 μM for 45 min incubation), as described[73]. ANG-2 fluorescence was excited at 495/10 nm wavelength and the emission was collected at 535/30 nm.

**Conditional NBCe1 knockdown in astrocytes**. To induce conditional NBCe1 knockdown in astrocytes, mice carrying a loxP-flanked NBCe1 allele (NBCe1$^{flox/flox}$)[74] were crossed with the mice expressing an inducible form of Cre (Cre$^{ERT2}$) under the astrocyte-specific GLAST promoter[75]. Tamoxifen (100 mg kg$^{-1}$) dissolved in corn oil was given to NBCe1$^{flox/flox}$:GLAST$^{CreERT2/+}$ mice (KO) at postnatal week 7 and the expression level of NBCe1 was examined 6 weeks after the tamoxifen treatment. Breeding was organised trough PCR genotyping obtained from tail DNA biopsies. First, we evaluated the cell-specificity of recombination by inducing the expression of the fluorescent reporter tdTomato in stop$^{fl/fl}$ tdTomato/GLAST-CreERT2 mice[75,76] following tamoxifen treatment at postnatal week 7. Astroglial recombination specificity of GLAST-CreERT2 mice has been reported in several prior studies[75,77]. Specific Cre-mediated recombination was confirmed in cortical protoplasmic astrocytes and Bergmann glia (cerebellum) by observing the characteristic pattern of tdTomato expression in the brains of stop$^{fl/fl}$ tdTomato /GLAST-CreERT2 mice. In animals used for the main experiment, genomic recombination of the NBCe1 locus was verified by PCR in various brain regions from NBCe1$^{flox/flox}$:GLAST$^{CreERT2/+}$ mice injected with tamoxifen and in two control groups including littermate NBCe1$^{flox/flox}$:GLAST$^{CreERT2/+}$ mice injected with vehicle (oil) and NBCe1$^{flox/flox}$:GLAST$^{CreERT2/-}$ mice injected with tamoxifen. Effective recombination was observed in the cortex, hippocampus, brainstem and cerebellum, but not in the kidneys of KO mice. No NBCe1 recombination was observed in tissues of control animals.

**Immunohistochemistry**. At the end of the experiments, the animals were given an anaesthetic overdose, the brains were removed, fixed in 4% paraformaldehyde and sliced (7 µm). The slices were first incubated in citrate buffer (10 mM citric acid) for 20 min and then in the blocking solution (3% bovine serum albumin, 10% normal goat serum, 0.1% Triton X-100) for 2 h followed by incubation with primary mouse monoclonal anti-NBCe1 (1:50; Santa Cruz Biotechnology, Cat#sc-515543, Lot#I2216) or mouse polyclonal anti-MAP2 antibodies (1:500; Sigma, Cat# M-1406) overnight at 4 °C. The sections were subsequently incubated in secondary anti-mouse antibody Dylight 488 (1:500; ThermoFisher, Cat #35502) for 2 h at room temperature.

Images of NBCe1 immunofluorescence in the cortices of NBCe1$^{flox/flox}$:GLAST$^{CreERT2/+}$ mice treated with vehicle (oil) or tamoxifen were acquired with identical optical settings using a confocal microscope (Olympus FV1000). Images were processed using ImageJ software (version 1.52 P). For each of the images, pixel intensity frequency distribution graph was generated, and pixel intensities were quantified by making binary images using the mean intensities of the two peaks of fluorescence. By measuring the area of the high-intensity pixel population, intensity of NBCe1 immunofluorescence in the cerebral cortex was compared between NBCe1$^{flox/flox}$:GLAST$^{CreERT2/+}$ mice treated with vehicle or tamoxifen.

**Genomic recombination and genotyping analysis**. Specificity of recombination was evaluated by PCR using DNA from the brain tissue of NBCe1$^{flox/flox}$:GLAST$^{CreERT2/+}$ mice treated with tamoxifen or vehicle using the following primers: recombination forward 5′-TGG TGG CTT AAA TTG CAAATGGC-3′; recombination reverse 5′-GCA GAT CCA CTC AGA GCT AAC T- 3′. PCR genotyping of mice carrying the floxed NBCe1, GLAST CreERT2 or ROSACAG −TdTomato/+alleles was performed using DNA from tail biopsies with the following primers: NBCe1 floxed forward primer: 5′ TGGTGGCTTAAATTGCAA ATGGC-3′; NBCe1 reverse primer: 5′-CATAACCCACTAAGTCCAGTACG-3′. GLAST CreERT2 forward primer 5′-GAGGCACTTGGCTAGGCTCTGAGGA-3′; GLAST-CreERT2 reverse primer 5′-GAGGAGATCCTGACCGATCAGTTGG-3′, and GLAST-CreERT2 CER1 primer 5-GGTGTACGGTCAGTAAATTGGACAT-3′. ROSACAG−TdTomato/+ wild-type forward primer 5′-AAG GGA GCT GCA GTG GAG TA-3′; ROSACAG −TdTomato/+ wild-type reverse primer 5′-CCG AAA ATC TGT GGG AAG TC-3′; ROSACAG−TdTomato/+ mutant forward primer 5′-GGC ATT AAA GCA GCG TAT CC-3′; ROSACAG−TdTomato/+ mutant reverse primer 5′-CTG TTC CTG TAC GGC ATG G-3′.

**RT qPCR**. Brain tissue from NBCe1$^{flox/flox}$:GLAST$^{CreERT2/+}$ mice treated with tamoxifen or vehicle were incubated in Trizol Reagent (Invitrogen) and RNA was extracted. Total RNA (2 µg) was reverse transcribed using the ImPrim-IITM Reverse Transcription System (Promega). qPCR was performed in triplicates using KAPA SYBR FASTA qPCR kit. NBCe1 expression was quantified using a comparative 2-ΔCt method[70] and presented as arbitrary units of expression, normalised to the expression of the cyclophilin gene. The primers used were the following: NBCe1_cDNA forward: 5′-GACAACATGCAGGGTGTGTTGGA-3′; NBCe1_cDNA reverse: 5′-CCACAGG CCAATCCAAAGACGA-3′; cyclophilin_cDNA forward: 5′- GGC AAT GCT GGA CCA AAC ACA A-3′; cyclophilin_cDNA reverse: 5-GTAAAATGCCCGCAAGTCA AAAG-3.

**Analysis of single-cell RNAseq data**. Single-cell RNA-sequencing (RNAseq) data from the mouse cerebral cortex were obtained from a publicly available database, collated and maintained by the Linnarsson group (Karolinska Institutet; http://mousebrain.org/). Cell dissociation, single-cell RNAseq and quality control methods are described in detail in the original report of the database[54]. Data processing and visualisation was performed using the Seurat package[78] (v.3.1.4) in R (v.3.6.0, "Planting of a Tree"). Code used for the analysis is available as part of the Supplementary Material. The combined mouse cortical cell RNAseq dataset was obtained from 50,478 cells with expression data for 27,998 genes. All cells that displayed nFeatures greater than 200, but less than 4000, and a percentage of mitochondrial RNA of less than 30% were included in the analysis. Remaining were 49,703 cells with average UMI counts (absolute number of observed transcripts; nCount) of 3124.92 and nFeatures (genes per cell) of 1592.39. At this point all the data were log normalized and scaled to 10,000 transcripts per cell. Find-VariableFeatures function[79] was used to identify the 4000 most variable genes between the cells to be used in principle component analysis (PCA). Prior to PCA, data were scaled with a linear transformation to ensure all genes were given equal weight in the subsequent analyses. Dimensional Reduction PCA was then performed on the scaled data up to and including the first 100 identified principle components. The Elbow method was used to determine the effective number of principle components; found to be 75. The K-nearest neighbour (KNN) graph was constructed using these 75 principle components. To cluster the cells, the Louvain method for community detection (Louvain algorithm) was used with resolution set to 2.0 as recommended for the datasets of this size[78]. Uniform Manifold Approximation and Projection (UMAP) was used to visualise the cell clusters in two dimensions based on the same 75 principle components used for clustering and yielded 63 distinct cell clusters. The distribution of the *Slc4a4* gene expression was then plotted across the identified clusters. Only four clusters showed average

scaled *Slc4a4* expression of >1.5 in more than 40% of the cells and were found to be grouped together after dimensionality reduction. These clusters (0, 9, 15 and 29) were scrutinised for the expression of characteristic cell-specific marker genes[80,81]. Gene expression data from these four identified clusters were then pooled and re-clustered by the same method described for the whole dataset with modifications. The KNN graph was constructed using 20 principle components. After visualisation by UMAP, 11 distinct astroglial clusters were identified. Seurat analysis was used to find differentially expressed genes between the identified cell clusters.

**Statistical analysis**. Imaging data were acquired and analysed using IQ3 imaging software (version 6.3; Andor, Oxford Instruments) or Olympus FluoView software (version 4; Olympus). CFM recordings in vivo were acquired using Power1401 interface and analysed offline using Spike2 software (version 7; Cambridge Electronic Design). Electrophysiological data from the in vitro recordings were acquired and analysed using pClamp 10.2 software. Statistical analysis of the data was performed using GraphPad-Prism software (version 8). Details of the statistical tests applied are provided within the figure legends. The data are reported as individual values and means ± SEM or as box-and-whisker plots.

**Reporting summary**. Further information on research design is available in the Nature Research Reporting Summary linked to this article.

## Data availability
The data that support the findings in this study are included within the Supplementary Material and available from the corresponding author upon request. The source data underlying Figs. 1a, b, d, e, 2d, 3f, h, 5c, e, 6b, g, h, and 7e, f, g and Supplementary Figs. 1a–d, 2f, g, 3b, d, f, 4b,d,f, h and 5a are provided as a Source Data file. Single-cell RNA-sequencing source data underlying Fig. 4 are available from a publicly available database (http://mousebrain.org/).

## Code availability
The computer code used for the analyses of the RNAseq data is included within the Supplementary Material.

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

## Acknowledgements

This work was supported by The Wellcome Trust (A.V.G. and D.A.R.) and the Fondecyt Iniciación Grant 11190678 (I.R.). D.A.R is a Wellcome Trust Principal Research Fellow (Ref: 212251). A.V.G is a Wellcome Trust Senior Research Fellow (Ref: 200893). CECs is funded by the Chilean Government through the Centers of Excellence Base Financing Program. We thank Héctor Oyarzún, Pablo Castro and Pamela Sanhueza (CECs) for technical assistance, Gary E. Shull (Cincinnati, USA), for providing NBCe1 flox mice, Frank Kirchhoff (Hamburg, Germany) for providing GLAST-CRE ERT2 mice and Hongkui Zeng (Seattle, USA) for providing Cre- reporter tdTomato mice. We are grateful to Professor Joachim W. Deitmer for his comments on an earlier version of the manuscript.

## Author contributions

A.V.G. conceived and directed the project; S.M.T., P.S.H., I.R., O.K., J.R.R. and P.Y.S. performed research; D.A.R. and O.K. designed the in vitro electrophysiological experiments. I.R. generated astrocyte-specific conditional NBCe1 knockout mice. L.F.B. contributed unpublished reagents/analytic tools; S.M.T., P.S.H., I.R., O.K. and J.R.R. analysed the experimental data; P.S.H. analysed the RNAseq data. A.V.G. and S.M.T. wrote the paper. All authors revised the article critically for important intellectual content.

## Competing interests

The authors declare no competing interests.
