## [Peer Review File · Nature Communications]

Manuscript ID: NCOMMS-19-35607
Responses to the referees' comments

We would like to thank all three reviewers and the Editors of *Nature Communications* for their time taken to evaluate our submission and overall positive assessment of our work. We are grateful for the detailed and mostly constructive comments provided and delighted to have an opportunity to re-submit our work. We now include additional experimental data requested by the reviewers, provide a full response to all the criticisms raised and submit a thoroughly revised manuscript. We believe very strongly that the results of the additional experimental studies and data analysis fully address all the criticisms raised and further support the conclusions reached.

Below we state the criticisms ("critique") and then provide our responses.

Reviewer #1:

This is a study on the role of astrocytes in general, and of the sodium-carbonate transporter NBCe1 in particular, in the control of brain extracellular pH (pHe). The experiments are based on a large variety of experimental approaches combined to the use of a novel mouse with conditional KO deletion of astrocytic NBCe1. However, there are several major issues that have to be considered in a future manuscript on this study.

Response: We would like to thank this reviewer for his/her time taken to review our manuscript. We respectfully disagree with the reviewer and strongly believe that the data obtained in our study conducted using *in vitro* and *in vivo* preparations provides the first integrative model of how extracellular pH is controlled in the mammalian brain.

First, the authors describe only a very small number of factors that influence brain pHe.

Response: Our study was not designed to study and describe ALL cellular and tissue mechanisms that influence brain extracellular pH. This would be an unreasonable expectation from any single research group. We aimed to investigate the role of astrocytes in maintaining extracellular pH homeostasis in face of changeable acid loads associated with variable levels of neuronal activity.

Second, the way pH dynamics and especially buffering has been dealt with is largely misleading and in part frankly erroneous.

Response: We believe that this comment may represent this reviewer's alternative view on the system. Respectfully, from the comments provided we were unable to envisage the reviewer's model of how extracellular pH homeostasis is achieved in the mammalian brain.

The multiple roles of brain pH-modulating neuronal signaling mechanisms and molecules (especially those of carbonic anhydrases) are not recognized by the authors.

Response: As mentioned above, our study was not designed to be exhaustive. Yet, we specifically mention the role of carbonic anhydrase in the text of the paper and in our graphical depiction of the proposed mechanism (Figure 8).

Critique: 1. In the beginning of the Introduction, out of a very large number of pH-modulatory targets which control neuronal excitability, the authors have chosen only two examples (refs 1 and 2; whereof the retina is not a very representative part of the CNS). Please add at least some refs on the pH sensitivity of ASICs, GABA_ARs, gap junctions, K⁺

channels, etc which will provide the reader an idea of the key role of pH in the activity-dependent modulation (not necessarily via local acidosis; see below) of neuronal functions.

Response: We thank the reviewer for this comment. We accept that there are many pH-sensitive processes in the CNS, however, in the Introduction we aimed to give the reader the two concise examples where neuronal activity can be affected by a failure of the pH control mechanisms. In the revised manuscript we now include additional references to other pH sensitive channels and receptors highlighted by the reviewer.

Critique: 2. The authors start the Results section by showing an experiment where applying acetazolamide (ATZ) i.p produces a fall in pHe. This experiment (which has been done before) is not easy to interpret, because ATZ is not very effective in diffusing across the BBB. ATZ would be first expected (a) to block CO₂ clearance from the blood, thereby leading to systemic hypercarbia. (b) While slowly diffusing across the BBB, the drug will affect both intra- and extracellular carbonic anhydrases (CAs) in this structure. Please note that the BBB is equipped with pH-regulatory carriers capable of generating and maintaining a pH gradient and therefore plays a role in setting pHe (see e.g. <https://www.ncbi.nlm.nih.gov/pubmed/19458287>). (c) When finally reaching the brain parenchyma, ATZ will first block the extracellular CAs (see review by Chesler which has been cited in the ms) and whether the drug does reach the intracellular compartments of neurons and glia within the time frame of an experiment of the present kind remains completely unknown. (d) Finally, a factor that may play a role in the effects of ATZ is the acidification of blood and brain stem, both of which are expected to influence respiration and thereby pCO₂ and pHe in a feedback manner. Thus, the summary in the text describing the ATZ data is highly superficial and lacks mechanistic information relevant for the present study "This indicates that brain cells continuously produce and extrude H⁺ into the extracellular space and that H⁺ buffering by HCO₃⁻ and carbonic anhydrase activity are critical for the maintenance of constant brain extracellular pH

Response: We thank the reviewer for this comment. Acetazolamide (ATZ) was applied intravenously (not i.p.) which would be expected to substantially reduce the time required for the drug to reach the brain parenchyma. While ATZ crosses BBB relatively poorly, it is a highly potent inhibitor of carbonic anhydrase and most certainly reaches neurons and glia within the time frame of the experiment described (several minutes). ATZ administered i.v. was shown to alter neuronal activity in the cortex within minutes (PMID:28087816). In our experiments the animals were paralyzed and mechanically ventilated, thus excluding the contribution of the potential effects of ATZ on breathing. To address this comment of the reviewer in the revision, we conducted additional experiments with application of ATZ directly in the vicinity of the recording electrode placed in the somatosensory cortex (Figure 1a). These experiments confirmed that inhibition of carbonic anhydrase with ATZ leads to extracellular acidification of the brain tissue (Figure 1a), supporting the underlying premise of the study.

Critique: This is echoed in the (obsolete and somewhat misleading summary Figure 1f) which has been shown again in almost the same form in Fig. 6. There are lots of relevant factors missing from this figure as is evident from my comments below, and it remains unclear whether the authors aim to describe the brain interstitial space in general or the neurovascular unit, which is a highly specialized compartment (see point 6 below).

Response: We believe that this comment represents this reviewer's point of view. If our model is 'obsolete' and 'misleading', then we would be greatly obliged if the reviewer could provide reference(s) to the contemporary view of the key mechanism underlying maintenance of extracellular pH in the mammalian brain and the *in vivo* data that support

such a model. Neurons within the brain are always closely apposed to the blood vessels (e.g. in the hippocampus neurons are within 8-25 μm from the nearest capillary and within 70-170 μm from the nearest arteriole; PMID: 10392829), therefore there is nothing special about the neurovascular unit. In a sense, the brain grey matter is a neurovascular unit, as depicted by our schematics.

Critique: 3. In light of the above shortcomings, it is obvious that pH buffering mechanisms in the brain have not been adequately described in the Introduction. Just a few points here: An equilibrium in a HCO_3/CO_2 buffer system is not attained "almost immediately" even with extracellular CAs around (no references provided in the ms), and this is particularly true for the brain interstitial space. This is because the local buffer system is far from open wrt CO_2 fluxes, as shown in <https://www.ncbi.nlm.nih.gov/pubmed/16611838>. Also, the buffering capacity of the HCO_3/CO_2 system is not primarily dependent on the "supply of HCO_3 " but rather on the rate-limiting factors of CO_2 movements (as in known from basic respiratory physiology).

Response: Carbonic anhydrase is one of the fastest mammalian enzymes, and by the expression 'almost immediately' we meant within a few seconds. We agree with the reviewer and in the revised manuscript modified this sentence to read: "*This equilibrium is rapidly attained by the activity of enzymes from the carbonic anhydrase family*". We agree that the CA activity is determined by the diffusion rate of its substrates. Yet, CO_2 is highly soluble and fast diffusing molecule. The cerebral perfusion rate is very high and removing acid as CO_2 , as depicted by our schematic (Figure 8), represents an efficient way of removing protons from the brain. The study described in the paper mentioned by the reviewer was performed in acute brain slices and at low temperature, devoid of blood flow and, therefore, may be poorly perfused deep in the slice as compared to the living brain. The level of extracellular carbonic anhydrase activity in acute slices may not represent the *in vivo* situations, as slicing procedure and superfusion with aCSF can lead to a significant loss of key extracellular enzymes.

Critique: 4. There is a clear semantic confusion wrt "buffering". The basics of physicochemical buffering are addressed above. However, the word "buffering" is also used in a manner analogous to K^+ buffering which obviously implies a physiological regulatory mechanism that dampens a change in ionic shifts. Thus the 2nd section in the Introduction should be rewritten, keeping in mind this distinction; and also what is said above, points 2-3. (The term "muffling" in ref 6 was coined by Roger Thomas in 1991 to clarify the terminology, but it has not been used in publications, with only four exceptions ever since.)

Response: We agree with the reviewer here, however, would prefer to use the term "buffering". It is derived from the verb 'to buffer' which has a meaning of 'to lessen or moderate the impact of (something)'. In this case extracellular acid loads. The term "buffering" is easily understood and widely used by the research community in similar contexts.

Critique: 5. How neuronal activity generates local (cf. Fig. 1f and 6) changes in pHe is based on a number of well-studied phenomena, where neuronal signaling and CA activities can have qualitatively different effects. A major point here is that pHe is not solely modulated by metabolism and presynaptically-released protons. Both glutamatergic and GABA_AR-mediated transmission lead to transmembrane fluxes of acid-base species which generate an *increase* in pHe. In the former case, the excitation-induced influx of Ca^{2+} has been shown to lead to a transient increase in pHe which is *exacerbated* by inhibitors of extracellular CAs (ICAes; see e.g. review by Chesler); while in the case of GABA_ARs, the efflux of HCO_3 leads to an increase in pHe which is *suppressed* by ICAes. Indeed, these two qualitatively

different mechanisms can be separated using selective recruitment of excitatory and inhibitory pathways with ICAs as analytic tools (<https://www.ncbi.nlm.nih.gov/pubmed/8793748>; and <https://www.ncbi.nlm.nih.gov/pubmed/8793748>).

The authors may appreciate the basic message here: neuronal activity does not necessarily lead to local acidosis; and CAs have roles other than simply buffering pH changes. These major points should be taken into account when considering the role and functions of astrocytes and NBCe1.

Response: We agree that in addition to the mechanisms mentioned and described in our study other processes may influence the brain extracellular pH. In the *in vitro* study mentioned by the reviewer the role of Ca²⁺ influx and alkaline shift was studied in conditions of complete glutamate and GABA receptor blockade (electrical stimulation in hippocampal slices at low temperature), where the physiological relevance of the data obtained may be questioned.

Neuronal signalling-related brain energy consumption was calculated to be equal to that of a human leg muscle running a marathon (PMID: 11598490). We hope that this reviewer would agree with the following fundamental facts: (1) Living cells must constantly extrude protons to maintain pH_i homeostasis because the membrane potential is more negative than the equilibrium potential for hydrogen ions (~ -12 mV); (2) protons are generated as a result of cell metabolism: i) anaerobic glycolysis and rapid turnover of glycolytically-derived ATP generates two protons per molecule of glucose, ii) aerobic metabolism leads to a net production of CO₂/H⁺ (PMID: 29461272, PMID: 6804190). Protons and CO₂ generated by active neurons and other brain cells must be removed if brain function is to be maintained. It is difficult to envisage the physiological significance of a mechanism that, in response to increases in neuronal activity, would promote intracellular accumulation of acid equivalents. Our conclusions are based on the data obtained *in vitro* and *in vivo* and allow us to propose an integrative model of homeostatic control of brain extracellular pH – to the best of our knowledge, the first of its kind.

Critique: 6. The title of the paper is “Active control of brain extracellular pH by astrocytes”. However, with practically no discussion on the structure and properties of the neurovascular unit (NVU), this compartment seems to be the focus of attention at least with reference to Figs. 1f and 7. Do the authors imply that all the acid-base alterations they detect are based on NVU properties and functions? If so, the title of the paper is misleading. And again, a number of problems will arise: (a) what is the fractional volume of NVUs in brain tissue? In the Introduction, a standard fractional volume of 20% has been assumed for the whole extracellular space, but it is clear that only an extremely small part of this is made of NVUs. (b) What evidence does this study present to support the idea that the NVUs comprise the main sources and sinks of H⁺ in the whole brain extracellular space, as assumed in Figs. 1f and 7? Activity-dependent acid-base shifts occur in ALL compartments of the brain interstitial tissue, even in white matter -- just to provide some food for further thoughts on these issues. (c) Given the highly specialized nature of the NVU, is it likely that data from glial cultures (e.g. Figs. 3 and 5) will provide solid information on acid-base homeostasis within the NVU?

Response: Respectfully we do not fully understand the relevance of this comment. We believe that this reviewer is using the term neurovascular unit (NVU) to refer to a narrow anatomical/structural concept of the space between a blood vessel and the enwrapping glial endfeet. At the Stroke Progress Review Group meeting of the National Institute of Neurological Disorders and Stroke (July 2001), the NUV was defined in much broader terms to encompass the anatomical and functional relationship between the brain cells (including neurons and glia) and the vasculature. As blood vessels and neurons are in close proximity

(e.g. in the hippocampus neurons are within 8-25 μm from the nearest capillary and within 70-170 μm from the nearest arteriole; PMID: 10392829), we would argue that the grey matter is almost ubiquitously a neurovascular unit. We agree that activity-dependent acid-base shifts occur in ALL compartments of the brain, but since all live cells are net producers of protons, these can only be removed from the brain by cerebral circulation as depicted by our schematics and by the mechanism we propose based on the data obtained (Figure 8). It is generally accepted that glial cultures allow studies of cellular mechanisms and complement the results obtained using reduced *in vitro* preparations (such as slices) and data obtained *in vivo*.

Critique: 7. The above considerations lead also to questions regarding the techniques of measuring pHe and also pH_i. (a) In the former case, it is not possible that the amperometric technique would have a spatial resolution to discern NVU-related pHe changes from those (most likely much larger) taking place outside the NVUs. (b) But even more worryingly, this kind of method will – by necessity – consume or produce H⁺ ions at the site of recording, thereby biasing the data. The cyclic nature of the technique does not necessarily safeguard against the above source of error, because pHe buffering is not necessarily similar for the two polarities. The currents used are large (nA levels) and the active surface area of the carbon fiber has not been provided. This technique is likely to be prone to changes in the H⁺ buffer capacity which should be tested in a calibration solution with varying pH levels and buffer capacities. None of these confounding factors have been discussed, and the Methods section plus the only reference cited (31) provide no information on the present amperometric technique.

Response: The voltammetry-based technique used for the recordings of extracellular pH in this study is described in detail in our recent paper published in *Biosensors and Bioelectronics* (doi.org/10.1016/j.biosx.2020.100034). The paper gives a detailed description of the technique, principles of H⁺ detection, calibration, specificity and interference with the detection of other analytes. The carbon fiber microelectrode measuring 7 μm in diameter and ~ 80 μm in length (active surface ~ 1800 μm^2) samples pH changes representative of that occurring within the current definition of the NVU. We did not aim to record from any specific microdomains.

This reviewer has made several assumptions regarding this technique that are not entirely accurate. First, this type of pH recording does not consume protons. Protons are not electroactive, therefore, cannot be oxidized or reduced and the detection strategy relies on modification of the electrode surface carbon by protons during the voltammetric scan. This reversible modification of the surface carbon groups by protons ensures that the electrode does not become a source or sink of protons and, thus, suitable for detection of rapid changes in pH. Second, while we agree with the reviewer that the polarity (+/-ve drive voltage) would affect the pH-dependent modification of carbon differently, the technique used in this study does not require polarity switching as all the recordings are performed with a negative voltage ramp.

Critique: (c) With regard to the calibrations of the *in vivo* BCECF signals, I have no idea how they were achieved. The only time calibration is dealt with is in relation to measurements of pH_i in glial cultures (line 501).

Response: BCECF signals *in vivo* were not calibrated as it is not possible to achieve with any degree of accuracy. Yet, the amplitudes of BCECF fluorescence changes in cortical astrocytes in response to increases in neuronal activity were similar to that induced by inhalation of 10% CO₂, indicative of intracellular acidification.

Critique: 8. Please provide an explanation for the observation that inhalation of 10% CO₂ produced an acidification in 60 out of 95 astrocytes in vivo (line 132). That pHi would remain constant in astrocytes during this kind of challenge is at odds with what has been published before on practically all types of cells, with few exceptions of specific cell types with CO₂-impermeant membranes.

Response: These data may suggest that some cortical astrocytes possess a mechanism that effectively counteracts CO₂-induced acidification of the intracellular compartment. This is an interesting question to address but is beyond the scope of this study.

Critique: 9. The conditional NBCe1 KO mouse is reported to have a suppression of only 25% of the protein (line 240). Given that "hypomorphs" (sometimes heterozygotes --/++) of other transmembrane ion carriers have hardly any detectable dysfunctions, it is amazing to learn that this kind of minor loss of NBCe1 leads to a situation where (lines 245-7) "extracellular pH was not maintained and a significant extracellular acidification developed in response to electrical stimulation of the forepaw". If this is so, one wonders what happens with the behavior (and pHe) of these mice before the experiments. With a striking qualitative alteration in activity-dependent pHe changes (as depicted in Fig. 6f), there must be a behavioral effect following tamoxifen treatment. More information on this conditional KO mouse is needed to put the data in Fig. 6f into a meaningful context.

Response: We are somewhat surprised by this reviewer's sceptical view of the data presented, given that global NBCe1 deficiency results in a lethal phenotype. This indicates the importance of this protein, and that the lack of NBCe1 function cannot be compensated by increased expression/function of any other transporter protein(s). Global NBCe1 knockout animals do not survive beyond the second postnatal week, therefore, it is not really surprising that the partial loss of NBCe1 we achieved in our model had such a profound effect on the mechanism that (in accord to all our other data) relies on NBCe1 activity.

Critique: 10. The heterogeneity of astroglial pHi responses in e.g. Fig. 1 is striking (see also in vitro data in Fig. 2), and raises some questions, including whether the hypothesized NVU-association is qualitatively heterogenous. What would this imply in terms of neurovascular coupling in general? Moreover, is it likely that BCECF-based recordings of pHi, where cell-soma responses must be dominating, would detect acid-base fluxes which are assumed to take place at the perivascular endfeet?

Response: We thank the reviewer for this comment. Indeed the heterogeneity of astroglial pHi responses is interesting and (as we discussed in the text of our original submission) is consistent with the evidence suggesting that forebrain astrocytes form a heterogeneous population in terms of their resting membrane potential with a larger subpopulation having membrane potential in the range between -90 to -70 mV and a 'depolarized' subpopulation with the membrane potential in the range between -60 to -30 mV. Considering that ion stoichiometry of astroglial NBCe1 is 1Na⁺:2HCO₃⁻ and the equilibrium potential of NBCe1 is between -68 to -74 mV in the majority of astrocytes this transporter would be expected to operate close to its reversal potential or in the outward mode and in a smaller subpopulation of astrocytes in the inward mode. To address this comment of the reviewer in the revision we conducted additional experiments and data analysis. Supporting the above hypothesis, we now report that ATP-evoked bicarbonate secretion and cytosolic acidification are markedly reduced when astrocytes are depolarized by raising the concentration of extracellular potassium (Supplementary Figure 3c,d). This means that when the astrocytic membrane potential is more positive than the equilibrium potential of NBCe1, outwardly directed transport of bicarbonate by NBCe1 is thermodynamically less favourable. In the revised manuscript we also report the results of our analysis of single cell RNAseq data of the mouse

cerebral cortex obtained from the publicly available database maintained by the Linnarsson group (PMID: 30096314). This analysis shows that (1) in the cortex, the expression of NBCe1 (*Slc4a4* gene) is confined almost exclusively to the astroglial population, and (2) astrocytes display heterogenous expression of both NBCe1 and carbonic anhydrase 2 (*Car2* gene) and form genetically distinct clusters of cells with high expression of both genes (Figure 4). Populations of astrocytes were identified by the expression of characteristic astrocyte-specific marker genes. These data suggest that the heterogeneity in astrocytic pHi responses recorded in the present study is likely to be due to variations in membrane potential and the level of expression of different pH regulatory proteins, including NBCe1.

Critique: 11. When stimulating the Schaffer collaterals (Fig 2), a large number of feed-forward and feedback interneurons are activated (see points 2-5). Looking at the bulky literature on GABA_A-mediated pH changes, these experiments which address the effects of glutamatergic excitatory transmission only should be done in the presence of GABA_A antagonists.

Response: We agree that the effects of glutamatergic excitatory transmission should be studied in the presence of GABA_A antagonists. However, we are not investigating the effects of glutamatergic excitatory transmission in isolation. Our goal is to understand the mechanisms underlying the maintenance of brain extracellular pH homeostasis at variable levels of neuronal activity including both excitatory and inhibitory synaptic interactions. In the opinion of the authors, the data obtained in the experiments involving GABA_A-receptor blockade would be difficult to interpret due to general disinhibition and stochastic changes in the activity of neuronal networks and associated variable changes in local pH.

Critique: 12. The idea that Fig. 7 would depict the "Brain CO₂/H⁺ cycle" (line 726) is an overstatement, to put it mildly. The scheme seems to depict some aspects of acid-base movements within the NVU and, even in this case, a large number of key molecules affecting pHi and pHe are missing (see comments above).

Response: As we argue above, our study was not designed to investigate and describe ALL cellular and tissue mechanisms that may influence brain extracellular pH. Yet, we strongly believe that our results obtained using *in vitro* and *in vivo* preparations provide strong evidence indicating the existence of a powerful astrocyte-driven mechanism that is critically important for the maintenance of brain extracellular pH homeostasis. The proposed mechanism is graphically depicted on this diagram. Unfortunately, from the comments provided by this reviewer we were unable to distil the reviewer's view of what is the key mechanism. If our model is 'obsolete' and 'misleading', then we would be greatly obliged if the reviewer could provide reference(s) to the contemporary view of the key mechanism underlying maintenance of extracellular pH in the mammalian brain and the *in vivo* data that support such a model.

Reviewer #2

Summary of findings:

- 1) pHo was stable vs neuronal activity *in vivo*.
- 2) Astrocytic pHi was decreased by neuronal stimulation both *in vivo* and *in vitro*
- 3) Astrocytic pHi activity-dependent changes (due to electrical stimulation or ATP/ADP application) were reduced by pharmacological or genetic blockage of NBCe1.
- 4) The induced pHi acidification and Ca²⁺_i increase was reduced by P2Y1 receptor blocker, PLC signaling inhibitor or inhibitor of Ca²⁺ release from internal store
- 5) pHo homeostasis collapsed by GLAST-specific knock-down of NBCe1.

Comments: However, as for any critical system, regulation of pHo is much more complex than currently envisioned by the authors. If one imagines a full-page diagram of systems that contribute to pHo, akin to “intermediary metabolism” charts, the authors have described a subset of interactions; there are many other influences on pHo. The point being that the conclusions appear to be far too wide-reaching compared to the experiments performed. The complexity of the system results in quite a few specific suggestions:

Response: We would like to thank this referee for his/her time taken to review our paper and overall positive assessment of our work. We fully agree that in addition to the mechanisms described in our study many other processes influence the extracellular pH. However, our study was not designed to investigate and describe ALL cellular and tissue mechanisms that may have an effect on brain extracellular pH. Understanding of the critical importance and high functional activity of the bicarbonate transporter NBCe1 emerged from several years of our research on pHi regulation in astrocytes. In the present study we aimed to put our knowledge of astrocytic NBCe1 function and regulation of bicarbonate transport into the context of extracellular pH control in the intact brain. We believe very strongly that our results obtained using *in vitro* and *in vivo* preparations provide strong evidence indicating the existence of a powerful astrocyte-driven mechanism that is critically important for the maintenance of brain extracellular pH homeostasis. Below we provide detailed responses to all the comments raised and discuss the results of the additional experiments requested by the reviewer.

Critique: 1. The paper needs to include evidence from the literature or experiments re: where and in what cell types NBCe1 is located. The location is critical to understanding the described experiments, particularly the pharmacological and ion-substitution experiments (which affect all NBCe1). a. What are the contributions of NBCe1s located in astrocytes vs neurons (principal cells vs interneurons) vs microglia vs perivascular cells?

Response: We thank the reviewer for this comment. NBCe1 is encoded by *Slc4a4* gene and was known previously to be highly expressed in astrocytes. Our earlier works suggested high functional expression of this transporter in cortical and brainstem astrocytes (PMID: 24453308, PMID: 25990710, PMID: 25820238, PMID: 27798130). Comparative analysis of brain transcriptome data published by Professor Barres (PMID: 18171944) suggested that NBCe1 is expressed exclusively in astrocytes. However, a study from Professor Chesler’s lab suggested that NBCe1 may also be expressed by hippocampal neurons (PMID: 21562261). There is also evidence that the brain endothelial cells may express NBCe1 (PMID: 27799072). To address this question of the reviewer in the revision we now include the results of our own comprehensive analysis of single cell RNAseq data of the mouse cerebral cortex obtained from the publicly available database maintained by the Linnarsson group (PMID: 30096314). The results of our analysis show that (1) in the cortex, the expression of NBCe1 (*Slc4a4* gene) is confined almost exclusively to the astroglial population, and (2) astrocytes display heterogeneous expression of both NBCe1 and carbonic anhydrase 2 (*Car2* gene) and form genetically distinct clusters of cells with high expression of both genes (Figure 4). Populations of astrocytes were identified by the expression of characteristic astrocyte-specific marker genes.

Critique: 2. What are the contributions of other membrane ion transport mechanisms?
a. HCO₃-permeable ligand-gated inhibitory ion channels such as GABA and glycine?
b. What are the contributions of Cl/HCO₃ exchangers, some of which are inhibited by the pharmacological antagonists directed at NBCe1 such as DIDS?
c. What are the contributions of proton/Na exchangers?
d. For 2a-c, on which cells are these ion channels, exchangers and transporters located?

Response: We thank the reviewer for this comment and fully agree that all these mechanisms may influence the brain extracellular pH. However, our study was not designed to study and describe ALL cellular and tissue mechanisms that may have an effect on brain extracellular pH. This would be an unreasonable expectation from a single research group. We focused on the mechanisms underlying signalling between neurons and astrocytes and aimed to investigate the role of astrocytes in maintaining brain extracellular pH homeostasis in face of changeable acid loads associated with variable levels of neuronal activity and metabolism.

a. We cannot completely exclude the contribution of GABA_A receptors or other anion channels in mediating bicarbonate release to the extracellular space. This would be governed by the extent of their recruitment during activity and the relative permeability of these channels to Cl⁻ vs HCO₃⁻. Studies of the potential role played by GABA_A receptors would require the experiments to be conducted in conditions of pharmacological GABA_A-receptor blockade. In the opinion of the authors, the data obtained in the experiments of this type would be difficult to interpret. GABA_A-receptor blockade would lead to general disinhibition and stochastic changes in the activity of neuronal networks with associated variable changes in local pH, unrelated to blockade of potential bicarbonate flux via GABA_A receptors.

b. It is generally believed that Cl⁻/HCO₃⁻ exchangers (AE's) predominantly function as acid loaders, responsible for restoring pH_i following intracellular alkalization. However, our previous studies (PMID: 25990710) provided strong evidence that pH_i regulation in astrocytes is independent of AE activity. Therefore, the effects of DIDS we report are most likely due to the inhibition of NBC. This conclusion is further corroborated by the data obtained with the use of a more specific NBC blocker S0859, as well as by the results of the experiments conducted in global and astrocyte-specific conditional NBCe1 knockout mice.

c. Sodium/hydrogen exchangers (NHE), particularly the NHE1 isoform, are ubiquitously expressed in all brain cells. As we illustrate on our schematic depiction of the proposed mechanism (Figure 8), NHEs extrude protons and contribute to the extracellular acid loads associated with increased neuronal activity. We now discuss these points in the revised manuscript.

Critique: 3. As would be expected for such a fundamentally important support system, there is substantial feedback. For example, astrocytic acidification increases purine release to the extracellular space independently of synaptic activity (Dulla et al. Neuron 2005, J Neurophysiol 2009).

Response: We thank the reviewer for this comment. We have also reported (astroglial) purine release in response to acidification in the respiratory centres of the brain (PMID: 16001070; PMID: 20647426). We agree with the reviewer that in the forebrain astrocytic acidification may facilitate the release of purines and may potentiate bicarbonate release (via the mechanism we describe) in a positive feedback manner and thus contribute to the maintenance of extracellular pH homeostasis. We now discuss this issue in the revised manuscript.

Critique: 4. The paper does not address *in vitro* vs *in vivo* changes in vascular perfusion. For example, activity-dependent astrocytic acidification may be more prominent *in vitro* because there is no blood flow to provide HCO₃⁻; instead, HCO₃⁻ is provided by the perfusate.

Response: We thank the reviewer for this comment. We believe that the strength of our study is that we test the underlying hypothesis using a combination of *in vitro* and *in vivo* experimental techniques. First, we characterize the process/phenomenon *in vivo*, then study the underlying cellular and molecular mechanisms using reduced preparations and then return to the *in vivo* models to determine whether the identified mechanism has functional

significance (addressed using genetic manipulation). We now briefly discuss the issue raised by the reviewer in the revised version of the manuscript.

Critique: 5. It is not clear why Na_i responses were enhanced in HCO_3^- -free media. The implied argument is that in the absence of NaHCO_3 , there is less export of Na_i , and this absence enhances the Na_i transient.

a. This argument in turn implies that in the presence of HCO_3^- , NaHCO_3 export rates exceed astrocytic NaKATPase flux. What is the evidence for this?

Response: Thank you for this question. We observed robust intracellular acidification in astrocytes in response to activation of P2Y_1 receptors. This acidification was abolished in bicarbonate-free medium or when NBCe1 activity was inhibited pharmacologically or in conditions of NBCe1 genetic deletion, suggesting that intracellular acidification is due to bicarbonate export via NBCe1. Bicarbonate efflux via NBCe1 would decrease intracellular sodium. $[\text{Na}^+]_i$ in all cells is regulated by multiple mechanisms involving activities of various ion channels and transporters. In response to ATP, the direction of $[\text{Na}^+]_i$ change would be determined by the NBCe1-mediated outward Na^+ transport, by Na^+ entry via enhanced sodium-calcium exchange (secondary to Ca^{2+} responses), potential activation of ionotropic P2X receptors, all occurring in parallel with the background activity of Na^+/K^+ ATPase. We recorded a net increase $[\text{Na}^+]_i$ in astrocytes in response to ATP (Supplementary Fig. S2e). In bicarbonate-free conditions, the amplitude of ATP-induced $[\text{Na}^+]_i$ responses was enhanced (Supplementary Fig. S2e,f). Because only the $\text{Na}^+/\text{HCO}_3^-$ cotransport is affected in the absence of HCO_3^- , this result is consistent with facilitation of Na^+ extrusion by outward NBCe1 activity. We now include a more detailed discussion of this issue in the revised manuscript.

Critique: b. What is the direction and contribution (to Na_i) of Na/H exchange in HCO_3^- -containing vs free media? One could imagine that when pH_o drops, this transporter might serve to reduce pH_o and H^+ and thereby.

Response: Thank you for this question. We did not study the role of Na^+/H^+ exchange (NHE) specifically, as the activity of NHEs expressed by all cells is generally believed to extrude protons and contribute to extracellular acid loads (counteracted by the mechanisms described in our study). NHEs exchange intracellular protons for extracellular sodium and thus contribute to cytosolic sodium load. The contribution of NHEs is illustrated on our schematic depiction of the proposed mechanism (Figure 8). We can only speculate that the NHE activity might increase as a result of intracellular acidification induced by P2Y_1 receptor activation.

Critique: 6. In vivo, pH_o was stable in the face of increased neuronal activity (increased stimulus frequency from 3 to 10 Hz, Supplementary Fig1a).

a) Is spike power (shown in Fig. 6h at 3 Hz stimulation) increased along with increment of stimulus frequency (i.e. is neuronal activity actually increased by this protocol)? If not, it may be a limitation of experimental design using anesthetized rat and the authors should reconsider the protocol (i.e., more stimulation frequency or prolonging the stimulation time, or recording in awake condition).

Response: We thank the reviewer for this comment and apologize for lack of clarity in our original submission. In our preliminary trials we found that 3 Hz is the optimal frequency of stimulation that induces the maximum spike power increases in the somatosensory cortex (please see the graph below). Stimulation frequencies higher than 3Hz evoke lower cortical activity due to the refractory properties of the somatosensory pathways/networks. In the revised submission we include the results of the experiments involving recordings of changes in extracellular pH in the somatosensory cortex evoked by 3Hz stimulations of different

durations (1, 3, 5, 10 and 20 s). It was found that pH_e was effectively maintained at different durations of stimulations. These data are now illustrated by the revised Supplementary Figure 1b.

Critique: b) Astrocytic pH_i acidification should be enhanced in an activity-dependent manner if bicarbonate was really supplied via NBCe1 'on demand' as authors claimed. These data are critical to the conclusions of the manuscript.

Response: We fully agree with the reviewer and our data reported in the original submission show that the degree of astroglial intracellular acidification is activity-dependent and mediated by NBC transporter activity (Figures 2 and 5 of the revised manuscript).

Critique: c) Neuronal activity-dependent collapse of pH_o is shown in mice with astrocyte specific knock-out of NBCe (Fig. 6).

Response: We believe that this is a critical piece of evidence in support of our proposed model of extracellular pH control. Even partial (25%) reduction of NBCe1 expression in astrocytes results in marked acidification of the extracellular space during periods of enhanced neuronal activity, suggesting that this transporter is critically important for the maintenance of extracellular pH homeostasis in the mammalian brain.

Critique: 7. The authors showed that astrocytes can transport bicarbonate bi-directionally depending on its membrane potential (Fig. 1d, Fig2d). I agree with the authors interpretation that astrocytic membrane potential is distributed around the equilibrium of NBCe (discussion, line 285-304). If so, effects of astrocytic membrane depolarization due to increased neuronal activity should be incorporated into the author's proposal of astrocytic pH_o homeostasis. The membrane depolarization should dampen the driving force for extruding bicarbonate via NBCe, which would limit 'on demand' bicarbonate supply. This limitation may be particularly important in pathophysiological conditions such as stroke or epilepsy. It would be valuable if authors can experimentally illustrate such a collapse of homeostatic regulation of pH_o by additional experiment with much higher stimulus intensity (or burst) on Fig 2d. The effect of membrane potential may explain the astrocytes that show no response or alkalization in an activity-dependent manner. This could be added to Fig 2d.

Response: We fully agree with the reviewer that astrocytic depolarization induced by increases in extracellular K^+ can modulate the direction of NBCe1 operation. In addition, our results strongly suggest that extracellular purines, also released as a result of enhanced synaptic activity, have a major impact on NBCe1 and facilitate the transport of bicarbonate. Therefore, the bicarbonate flux via NBCe1 appears to be determined by the net effect of

extracellular K^+ on membrane potential and activation of $P2Y_1$ receptors by the released purines. To address this point of the reviewer in the revised submission, we determined the effect of ATP on bicarbonate release and intracellular pH_i regulation at different extracellular potassium concentrations. It was found that raising the extracellular $[K^+]$ (from 3 mM to 7 mM and then to 20 mM) decreases the magnitude of ATP effect on intracellular pH in astrocytes (Supplementary Figure 3c,d). These data indicate that in astrocytes membrane depolarization reduces HCO_3^- release via NBCe1. These data are also in accord with the comment raised by the reviewer and suggest that under pathological conditions like epilepsy, stroke and spreading depolarization, when the extracellular $[K^+]$ increases significantly, the mechanism described in this study is likely to be disrupted and the maintenance of extracellular pH will be compromised as a result. We now discuss this issue in the revised manuscript.

Critique: 8. A limitation of this study is use of a non-ratiometric pH_i indicator. It is difficult to use a ratiometric dye in SR101-labeled astrocytes when using two-photon microscopy. But the estimation of pH_i is definitely less accurate compared to pH-sensitive fluorescent protein such as pHluorin. Authors used 10% CO_2 inhalation data as a positive control (supplementary Fig1. D, c), but the authors should add negative control data (stimulation with 0 mA intensity).

Response: We agree with the reviewer and now report the negative control data in the revised manuscript (Figure 1d).

Critique: 9. In the conditional knock-down experiment, the authors explain the lower degree of NBCe1 knock-down (25%, Fig. 6e) and mosaic expression pattern of this protein (Fig 6d) by the high functional expression and turnover rate of this protein. However, the expression pattern of reporter protein (tdTomato) itself also seems to be mosaic (fig 6c). Is this an issue with the efficacy of the AAV vector with GLAST promoter rather than the characteristics of the target protein?

Response: In these experiments we used tamoxifen-inducible Cre/loxP system to knock down the expression of NBCe1. This system was extensively used by many research groups due to the high cell specificity of the GLAST-CreERT2 in mouse astrocytes (Mori et al 2006, Saab et al 2012, Aida et al 2015, Supplie et al 2017, Jahn et al 2018). We did not use the AAVs. The observed mosaic pattern of expression is typical for this approach and we agree with the reviewer that our explanation provided in the original submission was incorrect. Ai14 mice express tdTomato fluorescence following Cre-mediated recombination (Madisen et al 2009). The expression of tdTomato was used in this work to confirm the cell specificity of recombination previously described in GLAST-positive cells, which varies between the brain regions (Jahn et al 2018). The pattern of tdTomato expression in the cortex was found to be mosaic, representing the heterogeneity in the level of GLAST expression in cortical astrocytes. In vehicle (oil) treated $NBCe1^{flox/flox}/GLAST^{CreERT2/+}$ mice homogeneous expression of NBCe1 was observed. The expression pattern of NBCe1 in $NBCe1^{flox/flox}/GLAST^{CreERT2/+}$ mice treated with tamoxifen was mosaic, similar to that observed in the recombination-reporter mice. We now revised this section of the manuscript to address this point of the reviewer. Thank you.

Critique: 10. A method for analyzing NBCe1 density (Fig. 6e) was not provided. Because quantification of immunostaining result is often tricky, author should state how to analyze the data clearly.

Response: We thank the reviewer for this comment and apologize for not including this information in the original submission. We now provide details of NBCe1 immunofluorescence density quantification in the revised manuscript.

Critique: 1. Abstract line 40-42, "In vivo and in vitro experiments..., up to 50% of all astrocytes release bicarbonate...". Because acidic portion in vivo is less than one-third (Fig. 1e), this notion is over-emphasized, I wonder.

Response: We thank the reviewer for this comment and apologize for this overstatement. We modified the text accordingly in the revised submission.

Critique: 2. Fig 6d lacks scale bars.

Response: Thank you for noticing this imperfection. We now provide scale bars for these images.

Critique: 3. BCECF and SR101 pictures in Fig 1e are not informative and not mentioned in main text or figure legend. For readers, it is unclear what cells were analyzed in Fig 1e.

Response: Thank you for this comment. We now modified these panels and provide a more detailed description of the data analysis with references to figure panels.

Critique: 4. Line 523, "GLASTCreERT2" should be "GLAST-CreERT2"

Response: Thank you for this comment, now corrected.

Critique: 5. The provider of BCECF is stated differently in line 424 (Invitrogen) and line 492 (Molecular Probes).

Response: Thank you for this comment, now corrected.

Reviewer #3

This is a very interesting study by an outstanding research team. They revisit an old idea regarding regulation of brain pH by showing that a subset of astrocytes buffer extracellular pH by releasing HCO_3^- via the NBC in response to a neural activity-induced acid load. These results are essentially opposite to a large body of evidence showing that neural activation results in extracellular acidification by HCO_3^- influx into astrocytes. Since regulation of brain pH is essential for normal neural function, this work is highly significant and the contradictory results are intriguing. Therefore, I think this work is potentially suitable for Nature Communication. My concerns are noted below.

Response: We would like to thank this referee for his/her time taken to review our paper and very positive assessment of our work. Below we provide detailed responses to all the comments raised, discuss the results of the additional experimental data requested by the reviewer and included in this revision, and submit a thoroughly revised manuscript.

Critique: 1) What is the H^+ sensitivity of the carbon fiber electrodes? How were these calibrated? Was it confirmed that acetazolamide did not interfere with the voltammetry signal? This is important because acetazolamide is electroactive and so depending on the electrode polarizing potential it may produce a current independent of H^+ (Gholivand and Parvin J. Electroanalytical Chem. 660:163-68, 2011).

Response: The voltammetry-based technique used for the recordings of extracellular pH in this study is described in detail in our recent paper published in *Biosensors and Bioelectronics* (doi.org/10.1016/j.biosx.2020.100034). The paper provides detailed description of the technique, principles of H⁺ detection, calibration, specificity and interference with the detection of other analytes. Regarding the sensitivity of the carbon fibre electrode to acetazolamide. Gholivand and Parvin described *electrooxidation* of acetazolamide. This process would require a positive drive voltage to accept electrons from the thiadiazole ring of the molecule. Our method of H⁺ detection is based on application of reducing (negative) waveform making electrooxidation of acetazolamide improbable.

Critique: 2) The authors should show that the electrochemical current produced by ATZ reaches a plateau and washes.

Response: Thank you. We found that it takes a very long time for extracellular pH to recover following systemic administration of ATZ. To address one of the comments raised by Reviewer 1, we repeated these experiments, but applied ATZ directly in the vicinity of the recording electrode placed in the somatosensory cortex. The results of these experiments show that inhibition of carbonic anhydrase with ATZ leads to a reversible extracellular acidification of the brain tissue. The new data are now illustrated by revised Figure 1a.

Critique: 3) 1st paragraph of results. The initial observation that blocking CA causes extracellular pH to increase is not novel and does not support the assertion that HCO₃⁻ is the only essential pHo buffer.

Response: We agree, but believe that showing the effect of acetazolamide is important to illustrate at the very beginning of this paper as it shows very clearly that living tissue constantly generates protons (that must be removed) and that inhibition of the system under investigation results in extracellular acidification. As mentioned above, for this revision we conducted additional experiments with application of ATZ directly in the vicinity of the pH recording electrode placed in the somatosensory cortex. Robust extracellular acidification induced by inhibition of carbonic anhydrase with ATZ strongly supports the premise of the study.

Critique: 4) In vitro, stimulating neural activity resulted in astrocyte acidification in many more cells than in vivo. Is this due to improved imaging in the slice or might the absence of blood flow in the slice contribute to this response?

Response: It would be difficult to answer this question directly as many factors may contribute to the differences observed between the results obtained using *in vivo* and *in vitro* preparations. In the brain the between-capillary distance is ~ 40 μm (PMID: 7317796) that ensures effective delivery of oxygen and nutrients as well as washout of metabolic waste products, such as CO₂. The slices are superfused from the sides and the differences observed may reflect the differences in tissue perfusion. We now discuss this issue in the revised manuscript.

Critique: 5) There is considerable evidence showing that stimulating neural activity including Schaffer collateral stimulation (PMID: 1380165) results in extracellular alkalosis (for reviews ref 3 of this ms or Ch 17 in pH and Brain function edited by Kaila and Ransom 1998). The work presented here suggests the opposite. To make this point more believable the authors confirm that they can replicate high freq stimulation induced extracellular acidification. Also, since Ba²⁺ inhibition of astrocyte Kir channels eliminated depolarization-

induced alkalization of astrocytes, it seems like the authors should confirm this and determine whether Ba²⁺ affects the the observed low stim induced increase in pHo.

Response: We thank the reviewer for this comment and agree that multiple mechanisms may influence brain extracellular pH. However, we feel very strongly that recordings of the extracellular pH in acute slices *in vitro* have significant limitations and may not accurately reflect *in vivo* situations due to lack of blood flow and open bath conditions. Our *in vivo* recordings in the somatosensory cortex demonstrated that during periods of enhanced neuronal activity, the extracellular pH remains stable or becomes slightly alkaline. In mice with astrocyte-specific NBCe1 knockdown this pH control mechanism is compromised and neuronal activation leads to extracellular acidosis.

Critique: 6) They also claim that under resting conditions the NBC moves HCO₃⁻ into the extracellular space. This also diverges from previous work. Therefore, they need better support of this possibility. First, what is the calculated NBC reversal potential in your system? The references used to suggest it's between -68 to -74 mV are fairly old and in the case of 34 not relevant to mouse. Second, show functionally that blocking the NBC under resting conditions increases outward current or hyperpolarizes membrane potential. Then repeat this when the ATP/ADP signaling mechanism is engaged and [Na⁺]_i is even higher.

Response: We thank the reviewer for raising this important question. Considering the average [Na⁺]_i of 15 mM, [HCO₃⁻]_i of 19 mM (pH_i ~ 7.3), [Na⁺]_o of 140 mM, [HCO₃⁻]_o of 26 mM and the 1Na⁺:2HCO₃⁻ stoichiometry, the calculated reversal potential of NBCe1 in a steady-state using the formula

$$E_{NBC} = \frac{RT}{zf} \frac{[Na^+]_o [HCO_3^-]_o^n}{[Na^+]_i [HCO_3^-]_i^n}$$

would be -71 mV. This suggests that an astrocyte with a membrane potential lower than E_{NBC} is poised to secrete bicarbonate via NBCe1. For this revision we conducted a series of electrophysiological experiments with the recordings of the membrane potential in a sample of hippocampal (CA1 region) astrocytes at resting conditions and under pharmacological NBC blockade with S0859. As predicted by the reviewer, NBC blockade hyperpolarized the membrane potential of hippocampal astrocytes (9 astrocytes/4 animals, median membrane potential gradually dropped from -80.7 mV to -85 mV during 10-min application of S0859, p = 0.0039; Supplementary Figure 3e,f), indicating that under resting conditions NBCe1 is operating in the outward mode.

Critique: 7) The evidence that HCO₃⁻ free medium potentiated purinergic dependent increases in [Na⁺]_i suggests that astrocytes in this preparation lack other Na⁺ handling mechanisms (e.g., Na/KATPase, EAAT, NHE to name a few). This is hard to believe and so should be tested.

Response: Thank you for this question. We observed robust intracellular acidification in astrocytes in response to activation of P2Y₁ receptors. This acidification was abolished in bicarbonate-free medium or when NBCe1 activity was inhibited pharmacologically or in conditions of NBCe1 genetic deletion, suggesting that intracellular acidification is due to bicarbonate export via NBCe1. Bicarbonate export via NBCe1 would decrease intracellular sodium. As pointed out by the reviewer, [Na⁺]_i in all cells is regulated by multiple mechanisms involving activities of various ion channels and transporters. In response to ATP, the direction of [Na⁺]_i change would be determined by the NBCe1-mediated outward Na⁺ transport, by Na⁺ entry via enhanced sodium-calcium exchange (secondary to Ca²⁺ responses), potential

activation of ionotropic P2X receptors, all occurring in parallel with the background activity of Na⁺/K⁺ ATPase. We recorded a net increase [Na⁺]_i in astrocytes in response to ATP (Supplementary Fig. S2e). In bicarbonate-free conditions, the amplitude of ATP-induced [Na⁺]_i responses was greatly enhanced (Supplementary Fig. S2e,f). Because only the Na⁺/HCO₃⁻ cotransport is altered in the absence of HCO₃⁻, this result is consistent with facilitation of Na⁺ extrusion by outward NBCe1 activity. We now include a more detailed discussion of this issue in the revised manuscript.

Critique: 8) P6L178-179 states that P2X contributes to the high [Na⁺]_i and so by de facto should influence HCO₃⁻ efflux and NCX activity, yet blocking P2X minimally affected ATP induced changes in astrocyte pH or Ca²⁺. These seem contradictory.

Response: Thank you for this comment and we apologize for the lack of clarity. We do not have any data to suggest that P2X receptors play a role in modulation of NBC activity in astrocytes. In the section of the text quoted by the reviewer, we only mentioned P2X receptors as one of the potential routes of Na⁺ entry. However, blockade of P2X receptors had no effect on ATP-induced intracellular pH changes suggesting that ionotropic ATP receptors are unlikely to be involved in the mechanisms described in our study. We now revised the text of the paper to make this point clear to the readers.

Critique: 9) Neural stimulation will presumably lead to glutamate uptake by astrocytes and this could cause intracellular acidification independent of HCO₃⁻. This should be controlled. Also, Schaffer collateral stimulation can activate inhibitory interneurons and since HCO₃⁻ can flux through GABA receptors, it seems like that should be controlled for.

Response: We thank the reviewer for this comment. In our experiments we observed that the activity-dependent intracellular pH changes in astrocytes are abolished by NBC blockade, strongly suggesting that the contribution of glutamate uptake mechanisms to intracellular acidification is minimal. Indeed, there is evidence that intracellular acidification induced by glutamate transporter activity is relatively small (and slow) due to the effective bicarbonate buffering (PMID: 8757252). The figure below illustrates D-aspartate-induced intracellular acidification in astrocytes recorded first in HEPES-buffered solution and then in CO₂/HCO₃⁻ (26 mM)-buffered medium.

We cannot completely exclude the contribution of GABA_A receptors or other anion channels in mediating bicarbonate release to the extracellular space. This would be governed by the extent of their recruitment during activity and the relative permeability of these channels to Cl⁻ vs HCO₃⁻. Studies of the potential role played by GABA_A receptors would require the experiments to be conducted in conditions of pharmacological GABA_A-receptor blockade. In

the opinion of the authors, the data obtained in the experiments of this type would be difficult to interpret. GABA_A-receptor blockade would lead to general disinhibition and stochastic changes in the activity of neuronal networks with associated variable changes in local pH, unrelated to blockade of potential bicarbonate flux.

Critique: 10) Does PLC modulation of the NBC involve direct PIP2 activation or indirect via IP₃/Ca²⁺? These are testable. It should be made clear that PLC modulation of the NBC increases activity but not directionality.

Response: In the *in vitro* heterologous expression system Thornell and colleagues (PMID: 22966160) showed that PIP2 dependent stimulation of NBCe1 variants B and C (the main variants expressed in astrocytes) is mediated by indirect action via IP₃ and endoplasmic calcium release. In the present study we observed a marked reduction of pH_i response to purinergic stimulation after depletion of intracellular calcium stores with thapsigargin (Figure 6f and Supplementary Figure 5a). These data suggest that the stimulation of NBCe1 activity in response to P2Y₁ receptor activation is mediated by IP₃ and calcium release from the intracellular stores. We now discuss this issue in more detail in the revised manuscript and make it clear that the activity, not directionality, of the transporter is likely to be modulated by purinergic signalling.

Critique: 11) There is considerable emphasis placed on astrocyte functional heterogeneity, yet no effort was made to identify mechanisms contributing to these differences. For example, do those astrocytes that show increased pH_i in response to neural activity lack P2Y₁R or otherwise do not show [Na⁺]_i or [Ca²⁺]_i changes?

Response: Both intracellular acidification (majority of astrocytes) and intracellular alkalinization (a proportion of astrocytes) in response to increases in local neuronal activity are blocked by pharmacological inhibition of NBC. These data suggest that all astrocytes express NBCe1 and (as we discuss in the text of the original submission) the directionality of the response is determined by the resting membrane potential of the cell. We thank the reviewer of this comment and for this revision performed additional experiments and data analysis. First, we determined the effect of ATP on bicarbonate release and intracellular pH_i regulation at different extracellular potassium concentrations. It was found that raising the extracellular [K⁺] (from 3 mM to 7 mM and then to 20 mM) decreases the magnitude of ATP effect on intracellular pH in astrocytes (Supplementary Figure 3c,d). These data suggest that in astrocytes membrane potential determines the efficacy of ATP-evoked outwardly directed HCO₃⁻ transport by NBCe1. Second, in the revised submission we now include the results of our analysis of single cell RNAseq data of the mouse cerebral cortex obtained from the publicly available database maintained by the Linnarsson group (PMID: 30096314). This analysis shows that (1) in the cortex, the expression of NBCe1 (*Slc4a4* gene) is confined almost exclusively to the astroglial population, and (2) astrocytes display heterogenous expression of both NBCe1 and carbonic anhydrase 2 (*Car2* gene) and form genetically distinct clusters of cells with high expression of both genes (Figure 4). Populations of astrocytes were identified by the expression of characteristic astrocyte-specific marker genes. Collectively, these data suggest that the variations in the membrane potential and probably the expression level of NBCe1 are responsible for heterogeneity of astroglial pH_i responses to increases in the neuronal activity.

Critique: 12) In the context of disease, neural activity induced extracellular acidification is considered to be a protective mechanism against hyperexcitability (H⁺ inhibits hippocampal neurons). Therefore, the mechanism proposed here seems pathological, neural activity-induced increase in pH_o would further amplify neural activity and favor seizures. Not protective as suggested in the discussion.

Response: We thank this reviewer for bringing this point to our attention. The mechanisms that maintain pH_o homeostasis addressed by the present study operate under normal physiological conditions. We always record stable or slightly alkaline extracellular pH changes in response to increases in the neuronal activity. pH-dependent neuronal inhibition or activation may be dependent on the type and the level of receptor/ion channel expression. For example, extracellular acidification would inhibit glutamate receptors and voltage gated calcium channels but would be expected to activate acid sensing ion channels, such as ASICs. Therefore, the net effect of extracellular acidification on the neuronal excitability will be governed by the balance of these influences. Under pathological conditions, like spreading depression, when extracellular $[K^+]$ rise is abnormally high, the operation of NBCe1 in the reversed mode would be expected to be reduced (Supplementary Figure 3c,d), thus contributing to extracellular acidosis. We now discuss these issues in the revised version of the manuscript.

Critique: 13) The authors compared mechanism described here to previous work in brainstem respiratory centers; however, they failed to note that in the brainstem the NBC moves HCO_3^- into astrocytes at the expense of pH_o . This is opposite to what they show in the CA1, this should be made clear to avoid confusion.

Response: Yes, we are fully aware of this as the reviewer mentions our own work (PMID: 27798130). The brainstem astrocytes are intermingled with the respiratory network that modulates breathing in accord with the arterial and brain pH/PCO_2 . The brainstem astrocytes are clearly distinct, reflecting the respiratory function of the region. We now discuss this important issue in more detail in the revised version of the manuscript. Thank you.